# U-Statistics for Importance-Weighted Variational Inference

**Javier Burroni**                                        *jburroni@cs.umass.edu*
*University of Massachusetts Amherst*

**Kenta Takatsu**[‡]                                       *ktakatsu@andrew.cmu.edu*
*Carnegie Mellon University*

**Justin Domke**                                          *domke@cs.umass.edu*
*University of Massachusetts Amherst*

**Daniel Sheldon**                                        *sheldon@cs.umass.edu*
*University of Massachusetts Amherst*

**Reviewed on OpenReview:** *https://openreview.net/forum?id=oXmwAPlbVw*

## Abstract

We propose the use of U-statistics to reduce variance for gradient estimation in importance-weighted variational inference. The key observation is that, given a base gradient estimator that requires $m > 1$ samples and a total of $n > m$ samples to be used for estimation, lower variance is achieved by averaging the base estimator on overlapping batches of size $m$ than disjoint batches, as currently done. We use classical U-statistic theory to analyze the variance reduction, and propose novel approximations with theoretical guarantees to ensure computational efficiency. We find empirically that U-statistic variance reduction can lead to modest to significant improvements in inference performance on a range of models, with little computational cost.

## 1 Introduction

An important recent development in variational inference (VI) is the use of ideas from Monte Carlo sampling to obtain tighter variational bounds (Burda et al., 2016; Maddison et al., 2017; Le et al., 2018; Naesseth et al., 2018; Domke & Sheldon, 2019). Burda et al. (2016) first introduced the *importance-weighted autoencoder* (IWAE), a deep generative model that uses the *importance-weighted evidence lower bound* (IW-ELBO) as its variational objective. The IW-ELBO uses $m$ samples from a proposal distribution to bound the log-likelihood more tightly than the conventional evidence lower bound (ELBO), which uses only 1 sample. Later, the IW-ELBO was also connected to obtaining better approximate posterior distributions for pure inference applications of VI (Cremer et al., 2017; Domke & Sheldon, 2018), or "IWVI". Similar connections were made for other variational bounds (Naesseth et al., 2018; Domke & Sheldon, 2019).

The IW-ELBO is attractive because, under certain assumptions [see Burda et al. (2016); Domke & Sheldon (2018)], it gives a tunable knob to make VI more accurate with more computation. The most obvious downside is the increased computational cost (up to a factor of $m$) to form a single estimate of the bound and its gradients. A more subtle tradeoff is that the signal-to-noise ratio of some gradient estimators degrades with $m$ (Rainforth et al., 2018), which makes stochastic optimization of the bound harder and might hurt overall inference performance.

To take advantage of the tighter bound while controlling variance, one can average over $r$ independent replicates of a base gradient estimator (Rainforth et al., 2018). This idea is often used in practice and requires a total of $n = rm$ samples from the proposal distribution.

---

[‡]Work done while at UMass.

Our main contribution is the observation that, whenever using $r > 1$ replicates, it is possible to reduce variance with little computational overhead using ideas from the theory of U-statistics. Specifically, instead of running the base estimator on $r$ independent batches of $m$ samples from the proposal distribution and averaging the result, using the same $n = rm$ samples we can run the estimator on $k > r$ *overlapping* batches of $m$ samples and average the result. In practice, the extra computation from using more batches is a small fraction of the time for model computations that are already required to be done for each of the $n$ samples. Specifically:

- We describe how to take an $m$-sample base estimator for the IW-ELBO or its gradient and reduce variance compared to averaging over $r$ replicates by forming a *complete U-statistic*, which averages the base estimator applied to every distinct batch of size $m$. This estimator has the lowest variance possible among estimators that average the base estimator over different batches, but it is usually not tractable in practice due to the very large number of distinct batches.

- We then show how to achieve most of the variance reduction with much less computation by using *incomplete U-statistics*, which average over a smaller number of overlapping batches. We introduce a novel way of selecting batches and prove that it attains a $(1 - 1/\ell)$ fraction of the possible variance reduction with $k = \ell r$ batches.

- As an alternative to incomplete U-statistics, we introduce novel and fast approximations for IW-ELBO complete U-statistics. The extra computational step compared to the standard estimator is a single sort of the $n$ input samples, which is very fast. We prove accuracy bounds and show the approximations perform very well, especially in earlier iterations of stochastic optimization.

- We demonstrate on a diverse set of inference problems that U-statistic-based variance reduction for the IW-ELBO either does not change, or leads to modest to significant gains in black-box VI performance, with no substantive downsides. We recommend always applying these techniques for black-box IWVI with $r > 1$.

- We empirically show that U-statistic-based estimators also reduce variance during IWAE training and lead to models with higher training objective values when used with either the standard gradient estimator or the doubly-reparameterized gradient (DReG) estimator (Tucker et al., 2018).

## 2 Importance-Weighted Variational Inference

Assume a target distribution $p(z, x)$ where $x \in \mathbb{R}^{d_x}$ is observed and $z \in \mathbb{R}^{d_z}$ is latent. VI uses the following evidence lower bound (ELBO), given approximating distribution $q_\phi$ with parameters $\phi \in \mathbb{R}^{d_\phi}$, to approximate $\ln p(x)$ (Saul et al., 1996; Blei et al., 2017):

$$\mathcal{L} = \mathbb{E}\left[\ln \frac{p(Z, x)}{q_\phi(Z)}\right] \leq \ln p(x), \qquad Z \sim q_\phi.$$

The inequality follows from Jensen's inequality and the fact that $\mathbb{E}\left[\frac{p(Z,x)}{q_\phi(Z)}\right] = p(x)$, that is, the *importance weight* $\frac{p(Z,x)}{q_\phi(Z)}$ is an unbiased estimate of $p(x)$.

Burda et al. (2016) first showed that a tighter bound can be obtained by using the average of $m$ importance weights within the logarithm. The importance-weighted ELBO (*IW-ELBO*) is

$$\mathcal{L}_m = \mathbb{E}\left[\ln \frac{1}{m} \sum_{i=1}^{m} \frac{p(Z_i, x)}{q_\phi(Z_i)}\right] \leq \ln p(x), \qquad Z_i \overset{\text{iid}}{\sim} q_\phi. \tag{1}$$

This bound again follows from Jensen's inequality and the fact that $\frac{1}{m}\sum_{i=1}^{m} \frac{p(Z_i,x)}{q_\phi(Z_i)}$, which is the sample average of $m$ unbiased estimates, remains unbiased for $p(x)$. Moreover, we expect Jensen's inequality to provide a tighter bound because the distribution of this sample average is more concentrated around $p(x)$ than the distribution of one estimate. Indeed, $\mathcal{L}_m \geq \mathcal{L}_{m'}$ for $m > m'$ and $\mathcal{L}_m \to \ln p(x)$ as $m \to \infty$ (Burda et al., 2016).

In importance-weighted VI (IWVI), the IW-ELBO $\mathcal{L}_m$ is maximized with respect to the variational parameters $\phi$ to obtain the tightest possible lower bound to $\ln p(x)$, which simultaneously finds an approximating distribution that is close in KL divergence to $p(z \,|\, x)$ (Domke & Sheldon, 2018). In practice, the IW-ELBO and its gradients are estimated by sampling within a stochastic optimization routine. It is convenient to define the *log-weight* random variables $V_i = \ln p(Z_i, x) - \ln q_\phi(Z_i)$ for $Z_i \sim q_\phi$ and rewrite the IW-ELBO as

$$\mathcal{L}_m = \mathbb{E}[h(V_{1:m})], \qquad h(v_{1:m}) = \ln \frac{1}{m} \sum_{i=1}^m e^{v_i}. \tag{2}$$

Then, an unbiased IW-ELBO estimate with $r$ replicates, using $n = rm$ i.i.d. log-weights $(V_{j,i})_{j=1,i=1}^{r,m}$ is

$$\hat{\mathcal{L}}_{n,m} = \frac{1}{r} \sum_{j=1}^r h(V_{j,1}, \ldots, V_{j,m}). \tag{3}$$

In $\hat{\mathcal{L}}_{n,m}$, we use the subscript $n$ to denote the total number of input samples used for estimation and $m$ for the number of arguments of $h$, which determines the IW-ELBO objective to be optimized.

For gradient estimation, an unbiased estimate for the IW-ELBO gradient $\nabla_\phi \mathcal{L}_m$ is:

$$\hat{\mathcal{G}}_{n,m} = \frac{1}{r} \sum_{j=1}^r g(Z_{j,1}, \ldots, Z_{j,m}), \qquad Z_{j,i} \overset{\text{iid}}{\sim} q_\phi, \tag{4}$$

where $g(z_{1:m})$ is any one of several unbiased "base" gradient estimators that operates on a batch of $m$ samples from $q_\phi$, including the reparameterization gradient estimator (Kingma & Welling, 2013; Rezende et al., 2014), the doubly-reparameterized gradient (DReG) estimator (Tucker et al., 2018), or the score function estimator (Fu, 2006; Kleijnen & Rubinstein, 1996).

## 2.1  IWVI Tradeoffs: Bias, Variance, and Computation

Past research has shown that by using a tighter variational bound, IWVI can improve both learning and inference performance, but also introduce tradeoffs such as those pointed out by Rainforth et al. (2018). In fact, there are several knobs to consider when using IWVI that control its bias, variance, and amount of computation. These tradeoffs can be complex so it is helpful to review the key elements as they relate to our setting, with the goal of understanding when and how IWVI can be helpful and providing self-contained evidence that the setting where U-statistics are beneficial can and does arise in practice.

Consider the task of maximizing an IW-ELBO objective $\mathcal{L}_m$ to obtain the tightest final bound on the log-likelihood. This requires estimating $\mathcal{L}_m$ and its gradient with respect to the variational parameters in each iteration of a stochastic optimization procedure. Assume there is a fixed budget of $n$ independent samples per iteration, where, for convenience, $n = rm$ for an integer $r \geq 1$, as above. The parameters $m$ and $r$ can be adjusted to control the estimation bias and variance at the cost of increased computation. Specifically:

- For a fixed $m$, by setting $r' > r$, we can reduce the variance of the estimator in Equation (4) by increasing the computational cost to $r'm > rm$ samples per iteration.

- For a fixed $r$, by setting $m' > m$ we can reduce the bias of the objective — that is, the gap in the bound $\mathcal{L}_{m'} \leq \ln p(x)$ — by increasing the computational cost to $rm' > rm$ samples per iteration.

  However, Rainforth et al. (2018) observed that increasing $m$ may also have the *negative* effect of worsening the signal-to-noise (SNR) ratio of gradient estimation (but also that this could be counterbalanced by increasing $r$). Later, Tucker et al. (2018) showed that, for the DReG gradient estimator, increasing $m$ can *increase* SNR; see also the paper by (Finke & Thiery, 2019) for a detailed discussion of these issues.

Overall, while the effect of increasing the number of replicates $r$ to reduce variance is quite clear, the effects of increasing $m$ are sufficiently complex that it is difficult to predict in advance when it will be beneficial.

However, an important premise of our work is that the optimal setting of $m$ is often strictly between 1 and $n$, since this is the setting where U-statistics can be used to reduce variance. To understand this, we can first reason from the perspective of a user that is willing to spend more computation to get a better model. Assuming the variational bound is not already tight, this user can increase $m$ as much as desired to tighten the bound, and then, increase $r$ as needed to control the gradient variance. This argument predicts that, for a sufficiently large computational budget and complex enough model (so that the bound is not already tight with $m = 1$), a value $m > 1$ will often be optimal.

From the perspective of a user with fixed computational budget, in which the number of optimization iterations is also being fixed, we could instead ask: "for a fixed $n$, what are the optimal choices of $m$ and $r = n/m$"? This question can be addressed empirically. Rainforth et al. (2018) reported in their Figure 6 that the extreme values, i.e., $m = 1$ or

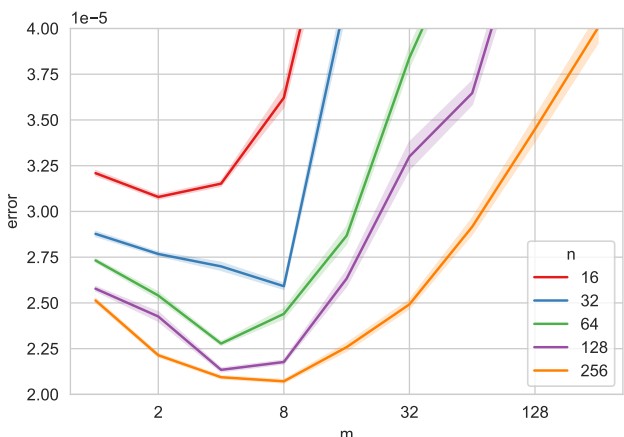

Figure 1: Distribution of the distance (error) between the distribution's covariance and that of the approximating distribution as a function of $m$ for different numbers of sampled points $n$, after training an approximating distribution using the standard IW-ELBO estimator. As we increase $n$, the optimal $m$ also increases, but at a slow rate. [See Section 6 for details.]

$m = n$, were never the best values. We found empirically that for some models, this result also holds for black box VI, i.e., the optimal choice of $m$ is strictly greater than 1 and less than $n$, as shown in Figure 1. See also Figure 5, which shows that similar observations apply when using the DReG estimator. In our analysis of 17 real Stan and UCI models, with $n = 16$, around half of them achieved the best performance for an intermediate value of $m$, depending on the approximating distribution and base gradient estimator [see Table 8 and 9 in Appendix G]. And we further conjecture that the fraction of real-world models with this property will increase as $n$ increases.

For the rest of this work we focus on methods that can reduce variance for the case when $1 < m < n$.

## 3 U-Statistic Estimators

We now introduce estimators for the IW-ELBO and its gradients based on U-statistics, and apply the theory of U-statistics to relate their variances. The theory of U-statistics was developed in a seminal work by Hoeffding (1948) and extends the theory of unbiased estimation introduced by Halmos (1946). For detailed background, see the original works or the books by Lee (1990) and van der Vaart (2000).

The standard estimators in Eqs. (3) and (4) average the base estimators $h(v_{1:m})$ and $g(z_{1:m})$ on disjoint batches of the input samples. The key insight of U-statistics is that variance can be reduced by averaging the base estimators on a larger number of overlapping sets of samples.

We will consider general IW-ELBO estimators of the form

$$\hat{\mathcal{L}}_{\mathcal{S}}(v_{1:n}) = \frac{1}{|\mathcal{S}|} \sum_{\mathbf{s} \in \mathcal{S}} h(v_{s_1}, \ldots, v_{s_m}), \tag{5}$$

where $\mathcal{S}$ is any non-empty collection of size-$m$ subsets of the indices $[\![n]\!] := \{1, \ldots, n\}$, and $s_i$ is the $i$th smallest index in the set $\mathbf{s} \in \mathcal{S}$. Since $\mathbb{E}\, h(V_{1:m}) = \mathcal{L}_m$, it is clear (by symmetry and linearity) that $\mathbb{E}\, \hat{\mathcal{L}}_{\mathcal{S}}(V_{1:n}) = \mathcal{L}_m$, that is, the estimator is unbiased. For now, we will call this a "U-statistic with kernel $h$", as it is clear the same construction can be generalized by replacing $h$ by any other symmetric function of $m$ variables[1], or "kernel", while preserving the expected value. Later, we will distinguish between different types of U-statistics based on the collection $\mathcal{S}$.

---

[1]Recall that a symmetric function is a function invariant under all permutations of its arguments.

We can form U-statistics for gradient estimators by using base gradient estimators as kernels. Let $g(z_{1:m})$ be any symmetric base estimator such that $\mathbb{E}\, g(Z_{1:m}) = \nabla_\phi \mathcal{L}_m$. The corresponding U-statistic is

$$\hat{\mathcal{G}}_{\mathcal{S}}(Z_{1:n}) = \frac{1}{|\mathcal{S}|} \sum_{\mathbf{s} \in \mathcal{S}} g(Z_{s_1}, \ldots, Z_{s_m}) \tag{6}$$

and satisfies $\mathbb{E}\, \hat{\mathcal{G}}_S(Z_{1:n}) = \nabla_\phi \mathcal{L}_m$.

### 3.1 Variance Comparison

How much variance reduction is possible for IWVI by using U-statistics? In this section, we first define the *standard IW-ELBO estimator* and *complete U-statistic IW-ELBO estimator*, and then relate their variances. For concreteness, we restrict our attention to IW-ELBO objective estimators, but analagous results hold for gradients by using a base gradient estimator as the kernel of the U-statistic.

We first express the standard IW-ELBO estimator $\hat{\mathcal{L}}_{n,m}$ in the terminology of Eq. (5):

**Estimator 1.** The *standard IW-ELBO estimator* $\hat{\mathcal{L}}_{n,m}$ of Eq. (3) is the U-statistic $\hat{\mathcal{L}}_{\mathcal{S}}$ formed by taking $\mathcal{S}$ to be a partition of $[\![n]\!]$ into disjoint sets, i.e., $\mathcal{S} = \{\{1, \ldots, m\}, \{m+1, \ldots, 2m\}, \ldots, \{(r-1)m+1, \ldots, rm\}\}$.

**Estimator 2.** The *complete U-statistic IW-ELBO estimator* $\hat{\mathcal{L}}_{n,m}^U$ is the U-statistic $\hat{\mathcal{L}}_{\mathcal{S}}$ with $\mathcal{S} = \binom{[\![n]\!]}{m}$, the set of all distinct subsets of $[\![n]\!]$ with exactly $m$ elements.

We will show that the variance of the $\hat{\mathcal{L}}_{n,m}^U$ is never more than that of $\hat{\mathcal{L}}_{n,m}$, and is strictly less under certain conditions (that occur in practice), using classical bounds on U-statistic variance due to Hoeffding (1948). Since $\hat{\mathcal{L}}_{n,m}^U$ is an average of terms, one for each $\mathbf{s} \in \binom{[\![n]\!]}{m}$, its variance depends on the covariances between pairs of terms for index sets $\mathbf{s}$ and $\mathbf{s}'$, which in turn depend on how many indices are shared by $\mathbf{s}$ and $\mathbf{s}'$. This motivates the following definition:

**Definition 3.1.** *Let $V_1, \ldots, V_{2m}$ be i.i.d. log-weights. For $0 \leq c \leq m$, take $\mathbf{s}, \mathbf{s}' \in \binom{[\![2m]\!]}{m}$ with $|\mathbf{s} \cap \mathbf{s}'| = c$. Using $h$ from* Eq. (2), *define*

$$\zeta_c = \mathrm{Cov}\Big[h(V_{s_1}, \ldots, V_{s_m}),\, h(V_{s_1'}, \ldots, V_{s_m'})\Big],$$

*which depends only on $c$ and not the particular $\mathbf{s}$ and $\mathbf{s}'$.*

In words, this is the covariance between two IW-ELBO estimates, each using one batch of $m$ i.i.d. log-weights, and where the two batches share $c$ log-weights in common. For example, when $m = 2$ we have

$$\zeta_0 = 0, \quad \zeta_1 = \mathrm{Cov}[\ln(\tfrac{1}{2}e^{V_1} + \tfrac{1}{2}e^{V_2}), \ln(\tfrac{1}{2}e^{V_1} + \tfrac{1}{2}e^{V_3})], \quad \text{and,} \quad \zeta_2 = \mathrm{Var}[\ln(\tfrac{1}{2}e^{V_1} + \tfrac{1}{2}e^{V_2})].$$

Then, due to Hoeffding's classical result,

**Proposition 3.2.** *With $\zeta_1$, and $\zeta_m$ defined as above, the standard IW-ELBO estimator $\hat{\mathcal{L}}_{n,m}$ (Estimator 1) and complete U-statistic estimator (Estimator 2) with $n = rm$ and $r \in \mathbb{N}$ satisfy*

$$\tfrac{m^2}{n}\zeta_1 \leq \mathrm{Var}[\hat{\mathcal{L}}_{n,m}^U] \leq \tfrac{m}{n}\zeta_m = \mathrm{Var}[\hat{\mathcal{L}}_{n,m}].$$

*Moreover, for a fixed $m$, the quantity $n\,\mathrm{Var}[\hat{\mathcal{L}}_{n,m}^U]$ tends to its lower bound $m^2\zeta_1$ as $n$ increases.*

*Proof.* The inequalities and asymptotic statement follow directly from Theorem 5.2 of Hoeffding (1948). The equality follows from the definition of $\zeta_m$. $\qquad\square$

Hoeffding proved that $m\zeta_1 \leq \zeta_m$. We observe in practice that there is a gap between the two variances that leads to practical gains for the complete U-statistic estimator in real VI problems.

A classical result of Halmos (1946) also shows that complete U-statistics are optimal in a certain sense: we describe how this result applies to estimator $\hat{\mathcal{L}}_{n,m}^U$ in Appendix B.

Finally, we conclude this discussion by stating the main analogue of Proposition 3.2 for gradient estimation. The result, also following from Theorem 5.2 of Hoeffding (1948), states that the complete U-statistic gradient estimator has total variance and expected squared norm no larger than that of the standard estimator:

**Proposition 3.3.** *Let $\hat{\mathcal{G}}_{n,m}$ and $\hat{\mathcal{G}}_{n,m}^U$ be the standard and complete-U-statistic gradient estimators formed using a symmetric base gradient estimator $g(z_{1:m})$ that is unbiased for $\nabla_\phi \mathcal{L}_m$ and the same index sets as $\hat{\mathcal{L}}_{n,m}$ and $\hat{\mathcal{L}}_{n,m}^U$, respectively. Then $\mathrm{tr}(\mathrm{Var}[\hat{\mathcal{G}}_{n,m}^U]) \leq \mathrm{tr}(\mathrm{Var}[\hat{\mathcal{G}}_{n,m}])$ and $\mathbb{E}\,\|\hat{\mathcal{G}}_{n,m}^U\|_2^2 \leq \mathbb{E}\,\|\hat{\mathcal{G}}_{n,m}\|_2^2$.*

We provide a proof in Appendix B.1.

## 3.2 Computational Complexity

There are two main factors to consider for the computational complexity of an IW-ELBO estimator:

1) The cost to compute $n$ log-weights $V_i = \ln p(Z_i, x) - \ln q(Z_i)$ for $i \in [\![n]\!]$, and

2) the cost to compute the estimator given the log-weights.

A problem with the complete U-statistics $\hat{\mathcal{L}}_{n,m}^U$ and $\hat{\mathcal{G}}_{n,m}^U$ is that they use $\left|\binom{[\![n]\!]}{m}\right| = \binom{n}{m}$ distinct subsets of indices in Step 2), which is expensive. It should be noted that these log-weight manipulations are very simple, while, for many probabilistic models, computing each log-weight is expensive, so, for modest $m$ and $n$, the computation may still be dominated by Step 1). However, for large enough $m$ and $n$, Step 2) is impractical.

# 4 Incomplete U-Statistic Estimators

In practice, we can achieve most of the variance reduction of the complete U-statistic with only modest computational cost by averaging over only $k \ll \binom{n}{m}$ subsets of indices selected in some way. Such an estimator is called an *incomplete U-statistic*. Incomplete U-statistics were introduced and studied by Blom (1976).

A general incomplete U-statistic for the IW-ELBO has the form in Eq. (5) where $\mathcal{S} \subsetneq \binom{[\![n]\!]}{m}$ is a collection of size-$m$ subsets of $[\![n]\!]$ that does not include every possible subset. We will also allow $\mathcal{S}$ to be a multi-set, so that the same subset may appear more than once. Note that the standard IW-ELBO estimator $\hat{\mathcal{L}}_{n,m}$ is itself an incomplete U-statistic, where the $k = r = \frac{n}{m}$ index sets are disjoint. We can improve on this by selecting $k > r$ sets.

**Estimator 3** (Random subsets)**.** The *random-subset incomplete-U-statistic estimator for the IW-ELBO* is the estimator $\hat{\mathcal{L}}_{\mathcal{S}_k}$ where $\mathcal{S}_k$ is a set of $k$ subsets $(\mathbf{s}_i)_{i=1}^k$ drawn uniformly at random (with replacement) from $\binom{[\![n]\!]}{m}$.

We next introduce a novel incomplete U-statistic, which is both very simple and enjoys strong theoretical properties.

**Estimator 4** (Permuted block)**.** The permuted block estimator is computed by repeating the standard IW-ELBO estimator $\ell$ times with randomly permuted log-weights and averaging the results. Formally, the *permuted-block incomplete-U-statistic estimator for the IW-ELBO* is the estimator $\hat{\mathcal{L}}_{\mathcal{S}_\Pi^\ell}$ with the collection $\mathcal{S}_\Pi^\ell$ defined as follows. Let $\pi$ denote a permutation of $[\![n]\!]$. Define $\mathcal{S}_\pi$ as the collection obtained by permuting indices according to $\pi$ and then dividing them into $r$ disjoint sets of size $m$. That is,

$$\mathcal{S}_\pi = \Big\{ \big\{\pi(1), \pi(2), \ldots, \pi(m)\big\}, \big\{\pi(m+1), \ldots, \pi(2m)\big\}, \ldots, \big\{\pi\big((r-1)m+1\big), \ldots, \pi(rm)\big\} \Big\}.$$

Now, let $\mathcal{S}_\Pi^\ell = \biguplus_{\pi \in \Pi} \mathcal{S}_\pi$ where $\Pi$ is a collection of $\ell$ random permutations and $\biguplus$ denotes union as a multiset. The total number of sets in $\mathcal{S}_\Pi^\ell$ is $k = r\ell$.

Both incomplete-U-statistic estimators can achieve variance reduction in practice for a large enough number of sets $k$, but the permuted block estimator has an advantage: its variance with $k$ subsets is never more than that of the random subset estimator with $k$ subsets, and never more than the variance of the standard

IW-ELBO estimator (and usually smaller). On the other hand, the variance of the random subset estimator is more than that of the standard estimator unless $k \geq k_0$ for some threshold $k_0 > r$.

**Proposition 4.1.** *Given $m$ and $n = rm$, the variances of these estimators satisfy the following partial ordering:*

$$
\underbrace{\operatorname{Var}[\hat{\mathcal{L}}_{n,m}^U]}_{\text{complete}} \overset{\text{(a)}}{\leq} \underbrace{\operatorname{Var}[\hat{\mathcal{L}}_{\mathcal{S}_\Pi^\ell}]}_{\text{permuted block}} \overset{\text{(b)}}{\underset{\text{(c)}}{\leq}} \overbrace{\operatorname{Var}[\hat{\mathcal{L}}_{n,m}]}^{\text{standard}} \underset{\text{random subset}}{\leq \underbrace{\operatorname{Var}[\hat{\mathcal{L}}_{\mathcal{S}_{r\ell}}]}} . \tag{7}
$$

*Moreover, if the number of permutations $\ell > 1$ and $\operatorname{Var}[\hat{\mathcal{L}}_{n,m}^U] < \operatorname{Var}[\hat{\mathcal{L}}_{n,m}]$, then* (b) *is strict; if $r = \frac{n}{m} > 1$, then* (c) *is strict. (Note that the permuted and random subset estimators both use $k = r\ell$ subsets.)*

*Proof.* By Def. 3.1, if $\mathbf{s}$ and $\mathbf{s}'$ are uniformly drawn from $\binom{[\![n]\!]}{m}$ and $\kappa = \left|\binom{[\![n]\!]}{m}\right|$, we have

$$
\mathbb{E}[\zeta_{|\mathbf{s} \cap \mathbf{s}'|}] = \sum_{\mathbf{s},\mathbf{s}' \in \binom{[\![n]\!]}{m}} \frac{\zeta_{|\mathbf{s} \cap \mathbf{s}'|}}{\kappa^2} = \sum_{\mathbf{s},\mathbf{s}' \in \binom{[\![n]\!]}{m}} \frac{\mathbb{E}[h(V_{s_1},\dots,V_{s_m})h(V_{s_1'},\dots,V_{s_m'})]}{\kappa^2} - \mathbb{E}[\hat{\mathcal{L}}_{n,m}^U]^2 = \operatorname{Var}[\hat{\mathcal{L}}_{n,m}^U]. \tag{8}
$$

Let $\pi_1,\dots,\pi_\ell$ be the random permutations. Observe that for $\mathbf{s},\mathbf{s}' \in \mathcal{S}_{\pi_i}$ distinct, i.e., two distinct sets within the $i$th block, $\mathbf{s}$ and $\mathbf{s}'$ are disjoint and then $h(V_{s_1},\dots,V_{s_m})$ is independent of $h(V_{s_1'},\dots,V_{s_m'})$. Hence, all dependencies between different sets are due to relations between permutations, i.e., each of the $\ell r$ terms will have a dependency with the $(\ell-1)r$ terms not in the same permutation. Therefore, it follows from (8) that the total variance of $\hat{\mathcal{L}}_{\mathcal{S}_\Pi^\ell}$ is

$$
\operatorname{Var}[\hat{\mathcal{L}}_{\mathcal{S}_\Pi^\ell}] = \tfrac{1}{\ell r}\zeta_m + (1 - \tfrac{1}{\ell})\operatorname{Var}[\hat{\mathcal{L}}_{n,m}^U], \tag{9}
$$

i.e., a convex combination of $\frac{1}{r}\zeta_m = \operatorname{Var}[\hat{\mathcal{L}}_{n,m}]$ and $\operatorname{Var}[\hat{\mathcal{L}}_{n,m}^U]$. Hence, using Proposition 3.2, (a) and (b) holds.

By a similar argument, the total variance of $\hat{\mathcal{L}}_{\mathcal{S}_{r\ell}}$ is

$$
\operatorname{Var}[\hat{\mathcal{L}}_{\mathcal{S}_{r\ell}}] = \tfrac{1}{\ell r}\zeta_m + (1 - \tfrac{1}{r\ell})\operatorname{Var}[\hat{\mathcal{L}}_{n,m}^U].
$$

Then, (c) holds because

$$
\operatorname{Var}[\hat{\mathcal{L}}_{\mathcal{S}_\Pi^\ell}] - \operatorname{Var}[\hat{\mathcal{L}}_{\mathcal{S}_{r\ell}}] = \tfrac{1}{\ell}(\tfrac{1}{r} - 1)\operatorname{Var}[\hat{\mathcal{L}}_{n,m}^U] \leq 0.
$$

$\square$

A remarkable property of the permuted-block estimator is that we can choose the number of permutations $\ell$ to guarantee what fraction of the variance reduction of the complete estimator we want to achieve. Say we would like to achieve 90% of the variance reduction; then it suffices to set $\ell = 10$. The following Proposition formalizes this result.

**Proposition 4.2.** *Given $m$ and $n = rm$, for $\ell \in \mathbb{N}$ the permuted-block estimator achieves a $(1 - 1/\ell)$ fraction of the variance reduction provided by the complete U-statistic IW-ELBO estimator, i.e.,*

$$
\underbrace{\operatorname{Var}[\hat{\mathcal{L}}_{n,m}]}_{\text{standard}} - \underbrace{\operatorname{Var}[\hat{\mathcal{L}}_{\mathcal{S}_\Pi^\ell}]}_{\text{permuted block}} = (1 - \tfrac{1}{\ell})(\underbrace{\operatorname{Var}[\hat{\mathcal{L}}_{n,m}]}_{\text{standard}} - \underbrace{\operatorname{Var}[\hat{\mathcal{L}}_{n,m}^U]}_{\text{complete}}).
$$

*Proof.* This follows directly from Eq. (9). $\square$

The conclusions of Propositions 4.1 and 4.2 do not depend on the kernel. This means they provide strong guarantees for our novel and simple permuted-block incomplete U-statistic with *any* kernel, which may be of general interest, and also imply the following result for gradients:

**Proposition 4.3.** *The conclusion of* Proposition 4.2 *holds with $\operatorname{Var}[\hat{\mathcal{L}}]$ replaced by either $\mathbb{E}[\|\hat{\mathcal{G}}\|_2^2]$ or $\operatorname{tr}([\operatorname{Var}[\hat{\mathcal{G}}])$, for each pair $(\hat{\mathcal{L}}, \hat{\mathcal{G}})$ of objective estimator and gradient estimator that use the same collection $\mathcal{S}$ of index sets, and for any base gradient estimator $g(v_{1:m})$.*

## 5   Efficient Lower Bounds

In the last section, we approximated the complete U-statistic by averaging over $k \ll \binom{n}{m}$ subsets. For example, by Proposition 4.2, we could achieve 90% of the variance reduction with $10\times$ more batches than the standard estimator, and the extra running-time cost is often very small in practice. An even faster alternative is to approximate the kernel in such a way that we can compute the complete U-statistic without iterating over subsets. In this section, we introduce such an approximation for the IW-ELBO objective, where the extra running-time cost is a single sort of the $n$ log-weights, which is extremely fast. Furthermore, Proposition 5.2 below will show that it is always a lower bound to $\hat{\mathcal{L}}^U_{n,m}$ and has bounded approximation error, so its expectation lower bounds $\mathcal{L}_m$ and $\ln p(x)$; thus, it can be used as a surrogate objective within VI that behaves well under maximization. We then introduce a "second-order" lower bound, which has provably lower error. Unlike the last two sections, these approximations do not have analogues for arbitrary gradient estimators such as DReG or score function estimators. For optimization, we use reparameterization gradients of the *surrogate* objective.

**Estimator 5.** The *approximate complete U-statistic IW-ELBO estimator* is

$$\hat{\mathcal{L}}^{\mathcal{A}}_{n,m}(V_{1:n}) = \binom{n}{m}^{-1} \sum_{\mathbf{s} \in \binom{[\![n]\!]}{m}} \max(V_{s_1}, \ldots, V_{s_m}) - \ln m.$$

This estimator uses the approximation $\ln \sum_{i=1}^{m} e^{v_i} \approx \max\{v_1, \ldots, v_m\}$ for log-sum-exp. The following Proposition shows that we can compute $\hat{\mathcal{L}}^{\mathcal{A}}_{n,m}$ *exactly* without going over the $\binom{n}{m}$ subsets but instead taking only $O(n \ln n)$ time. The intuition is that each of the $n$ log-weights will be a maximum element of some number of size-$m$ subsets, and each such term in the summation for $\hat{\mathcal{L}}^{\mathcal{A}}_{n,m}$ will be the same. Moreover, we can reason in advance how many times each log-weight will be a maximum.

**Proposition 5.1.** *For any $v_{1:n} \in \mathbb{R}^n$, it holds that*

$$\hat{\mathcal{L}}^{\mathcal{A}}_{n,m}(v_{1:n}) \equiv \binom{n}{m}^{-1} \sum_{i=1}^{n} b_i v_{[i]} - \ln m,$$

*where $b_i = \binom{n-i}{m-1}$, if $i \in [\![n-(m-1)]\!]$ (and 0 otherwise), and $[\cdot]: [\![n]\!] \to [\![n]\!]$ is a permutation s.t., the sequence of log-weights $v_{[1]}, \ldots, v_{[n]}$ is non-increasing.*

*Proof.* For $\mathbf{s} \in \binom{[\![n]\!]}{m}$, let $\mathbf{v_s} = (v_{s_1}, \ldots, v_{s_m})$. We can see that $\max \mathbf{v_s} = v_{[i]}$ where $i$ is the smallest index in $\mathbf{s}$. Thus,

$$\sum_{\mathbf{s} \in \binom{[\![n]\!]}{m}} \max \mathbf{v_s} = \sum_{i=1}^{n} b_i v_{[i]},$$

where $b_i$ is the number of sets $\mathbf{s} \in \binom{[\![n]\!]}{m}$ with minimum index equal to $i$. The conclusion follows because there are $n - i$ indices larger than $i$, but we can take $m - 1$ of them only when $i \in [\![n - (m-1)]\!]$.   $\square$

To further understand both the computational simplification and the quality of this approximation, consider this real example of computing the (non-approximate) complete U-statistic IW-ELBO estimator $\hat{\mathcal{L}}^U_{4,2}$. Suppose that the sampled log-weights are

$$\mathbf{v} = (-6034.091, -4351.335, -4157.236, -5419.201).$$

Given the $\binom{4}{2}$ sets, we can evaluate the kernel $h(v_i, v_j) = \ln(e^{v_i} + e^{v_j}) - \ln 2$ on each of them to generate the following table:

| $(v_i, v_j)$ | $h(v_i, v_j)$ |
|---|---|
| $(-6034.091, -4351.335)$ | $-4352.028$ |
| $(-6034.091, -4157.236)$ | $-4157.930$ |
| $(-6034.091, -5419.201)$ | $-5419.895$ |
| $(-4351.335, -4157.236)$ | $-4157.930$ |
| $(-4351.335, -5419.201)$ | $-4352.028$ |
| $(-4157.236, -5419.201)$ | $-4157.930$ |
| Mean | $-4432.956$ |

At three decimal points of precision, we see that $h(v_i, v_j) = \max(v_i, v_j) + \ln 2$ and therefore $-4157.930$, $-4352.028$, and $-5419.895$ each appear $\binom{3}{1}$ times, $\binom{2}{1}$ times, and once, respectively.

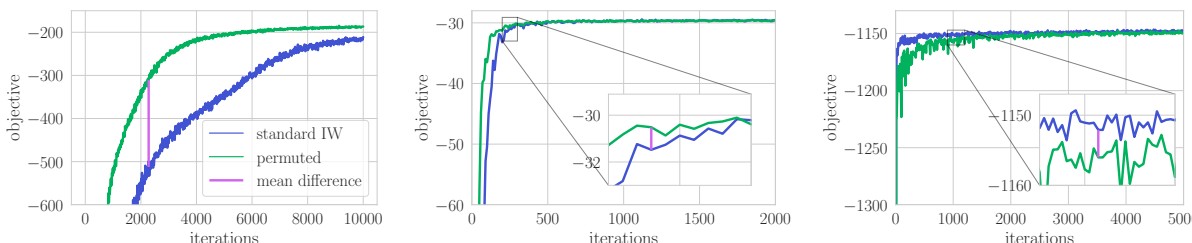

Figure 2: Median envelope of the objective using the permuted-block and standard IW-ELBO estimators for the `mushrooms` (left), `mesquite` (center) and `electric-one-pred` (right) models. In all cases we used $n = 16$ and $m = 8$. For reference there is a line segment of length similar to the average objective difference, respectively, 202.08, 0.97, and $-3.91$.

## 5.1 Accuracy and Properties of the Approximation

It is straightforward to derive both upper and lower bounds of the complete U-statistic IW-ELBO estimator $\hat{\mathcal{L}}_{n,m}^U$ from this approximation.

**Proposition 5.2.** *For any set of log-weights $v_{1:n} \in \mathbb{R}^n$, it holds that*

$$\hat{\mathcal{L}}_{n,m}^{\mathcal{A}}(v_{1:n}) \leq \hat{\mathcal{L}}_{n,m}^U(v_{1:n}) \leq \hat{\mathcal{L}}_{n,m}^{\mathcal{A}}(v_{1:n}) + \ln m. \tag{10}$$

*Moreover, the first inequality is strict unless $m = 1$. On the other hand, the second inequality is an equality when all log-weights are equal.*

*Proof.* This is a direct application of well-known inequalities for log-sum-exp. Let $h(v_1, \ldots, v_m) = \ln \sum_{i=1}^M e^{v_i}$ and $f(v_1, \ldots, v_m) = \max\{v_1, \ldots, v_m\}$. Then, for all $v_{1:m} \in \mathbb{R}^m$,

$$f(v_{1:m}) \leq h(v_{1:m}) \leq f(v_{1:m}) + \ln m. \tag{11}$$

To see this, write $\hat{v} = \max\{v_1, \ldots, v_m\}$. Then,

$$\tfrac{1}{m} e^{\hat{v}} \leq \tfrac{1}{m} \sum_{j=1}^m e^{v_j} \leq e^{\hat{v}}. \tag{12}$$

Eq. (11) follows from applying $\ln$ to (12). Eq. (10) then follows from (11) and the definitions of $\hat{\mathcal{L}}_{n,m}^{\mathcal{A}}$ and $\hat{\mathcal{L}}_{n,m}^U$. $\square$

One comment about the approximation quality is in order: in the limit as the variance of the log-weights decreases, the second inequality in the bounds above becomes tight, and the approximation error of $\hat{\mathcal{L}}_{n,m}^{\mathcal{A}}(v_{1:n})$ approaches its maximum $\ln m$. This can be seen during optimization when maximizing the IW-ELBO, which tends to reduce log-weight variance [cf. Figure 4].

## 5.2 Second-Order Approximation

Based on our understanding of the approximation properties of $\hat{\mathcal{L}}_{n,m}^{\mathcal{A}}$, we can add a correction term to obtain a second-order approximation.

**Estimator 6.** For $2 \leq m \leq n$, the *second-order approximate complete-U-statistic IW-ELBO estimator* is

$$\hat{\mathcal{L}}_{n,m}^{\mathcal{A},2}(V_{1:n}) = \hat{\mathcal{L}}_{n,m}^{\mathcal{A}}(V_{1:n}) + \binom{n}{m}^{-1} \sum_{i=1}^{n-(m-1)} \tilde{b}_i \ln(1 + e^{\Delta V_{[i]}}), \tag{13}$$

where $\Delta V_{[i]} = V_{[i+1]} - V_{[i]}$ and $\tilde{b}_i = \binom{n-1-i}{m-2}$.

This can still be computed in $O(n \ln n)$ time and gives a tighter approximation than $\hat{\mathcal{L}}_{n,m}^{\mathcal{A}}$.

**Proposition 5.3.** *For all $v_{1:n} \in \mathbb{R}^n$,*

$$\hat{\mathcal{L}}_{n,m}^{\mathcal{A}}(v_{1:n}) < \hat{\mathcal{L}}_{n,m}^{\mathcal{A},2}(v_{1:n}) \leq \hat{\mathcal{L}}_{n,m}^{U}(v_{1:n}).$$

*Moreover, the second inequality is an equality exactly when $m = n = 2$.*

*Proof.* The first inequality follows directly because the terms in the summation of (13) are positive reals.

For the second inequality, take $\mathbf{s} \in \binom{[\![n]\!]}{m}$ and let $i$ be the smallest index in $\mathbf{s}$. If $\mathbf{s}$ is one of the $\binom{n-1-i}{m-2}$ sets on which $i$ is the smallest index and $i + 1 \in \mathbf{s}$, then

$$\frac{1}{m} e^{v_{[i]}} \left(1 + e^{v_{[i+1]} - v_{[i]}}\right) = \frac{e^{v_{[i]}} + e^{v_{[i+1]}}}{m} \leq \frac{1}{m} \sum_{s \in \mathbf{s}} e^{v_s}.$$

If $i + 1 \notin \mathbf{s}$, we know that $\frac{1}{m} e^{v_{[i]}} \leq \frac{1}{m} \sum_{j=1}^{m} e^{v_{s_j}}$. We finish by applying logarithm to both inequalities and the definition of $\hat{\mathcal{L}}_{n,m}^{\mathcal{A}}$ and $\hat{\mathcal{L}}_{n,m}^{U}$. □

In contrast to $\hat{\mathcal{L}}_{n,m}^{\mathcal{A}}$, the second-order approximation is not a U-statistic. However, it is a tighter lower-bound of $\hat{\mathcal{L}}_{n,m}^{U}$.

**Note 5.4.** To use the approximations as an objective, we need them to be differentiable. If the distribution of $W$ is absolutely continuous, then the approximations are almost surely differentiable because `sort` is almost surely differentiable, with Jacobian given by the permutation matrix it represents [cf. Blondel et al. (2020)].

## 6 Experiments

In this section, we empirically analyze the methods proposed in this paper. We do so in three parts: we first study the gradient variance, VI performance, and running time for IWVI in the "black-box" setting[2]; we then focus on a case where the posterior has a closed-form solution, using random Dirichlet distributions; and finally, we study the performance of the estimators for Importance-Weighted Autoencoders.

For black-box IWVI, we experiment with two kinds of models: Bayesian logistic regression with 5 different UCI datasets (Dua & Graff, 2017) using both diagonal and full covariance Gaussian variational distributions,[3] and a suite of 12 statistical models from the Stan example models (Stan Development Team, 2021; Carpenter et al., 2017), with both diagonal (all models) and full covariance Gaussian (10 models[4]) approximating

---

[2]That is, VI that uses only black-box access to $\ln p(z, x)$ and its gradients.

[3]That is, $p(y \,|\, \theta) = \prod_{i=1}^{N} \text{Bernoulli}\big(y_i; \text{logistic}(\theta^T x_i)\big)$ for fixed $x_i \in \mathbb{R}^d$ and $p(\theta) = \mathcal{N}(\theta; \mathbf{0}, \sigma^2 \mathbf{I}_d)$, and $V = \ln p(\theta, y) - \ln q(\theta)$ for $\theta \sim q(\theta)$ with either $q(\theta) = \mathcal{N}(\theta; \mu, \text{diag}(w))$ or $q(\theta) = \mathcal{N}(\theta; \mu, LL^T)$; we optimize over $(\mu, w)$ or $(\mu, L)$, with $w$ constrained to be positive (via exponential transformation) and $L$ constrained to be lower triangular with positive diagonal (via softplus transformation). Parameters were randomly initialized prior to transformations from iid standard Gaussians.

[4]The `irt-multilevel` model diverged for all configurations using a full covariance Gaussian.

distributions. We provide additional information regarding the models in Appendix C. For each model, the variational parameters were optimized using stochastic gradient descent with fixed learning rate for 15 different logarithmically spaced learning rates. We used $n = 16$ samples per iteration except for the running time analysis, and experimented with $m \in \{2, 4, 8\}$. Since this is a stochastic optimization problem, we ran every combination of model, learning rate, $n$, and $m$, using 50 different random seeds to assess typical performance. We used the reparameterization gradient estimator as the base gradient estimator, and also provide in Appendix D and G (very similar) results for the doubly-reparameterized (DReG) gradient estimator.

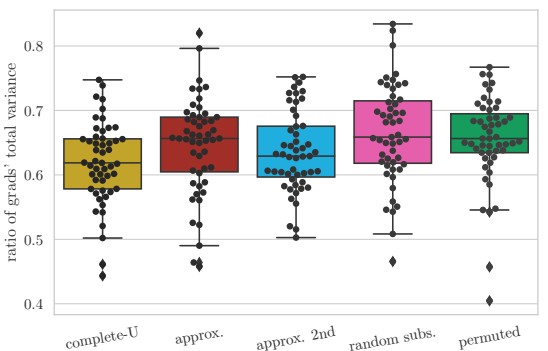
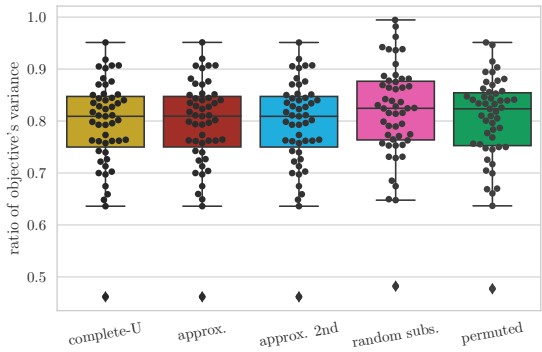

(a) Ratio of gradient's total variance.    (b) Ratio of the objective's variances.

Figure 3: Ratios of the trace of the variance (i.e., the total variance) of different proposed gradient estimators to that of the standard gradient estimator, and objective's variances (b) for the `mushrooms` dataset ($d = 96$). All ratios are below 1, which indicates variance reduction. The estimators can be ordered by variance: the complete-U-statistic estimator and second-order approximation are lowest, followed by the permuted-block and first-order approximation, and finally the random subsets estimator. Since $\ell = 20$, we expect the permuted-block estimator to achieve $1 - \frac{1}{20} = 0.95$ of the variance reduction; the estimated variance reduction is 91.24% and 95.72% for the objective.

**Gradient Variance** We first confirm empirically that U-statistics reduce the variance of gradients within IWVI. For each random seed, we performed IWVI using the complete U-statistic $\hat{\mathcal{L}}_{n,m}^{U}$ for 10,000 iterations. Every 200 iterations, we computed the gradients, given the values of the parameters at that time, for each of the alternative gradient estimators: the standard estimator, the complete U-statistic estimator, its approximations, the permuted-block estimator with $\ell = 20$, and the random subsets estimator with $k = 20\frac{n}{m}$ (a number of sets equal to the permuted version). In all cases we used $n = 16$ and $m = 8$. For each gradient estimator $\hat{\mathcal{G}}$, we estimate the total variance $\text{tr}(\text{Var}[\hat{\mathcal{G}}])$ using 200 independent gradient samples.

Figure 3–(a) shows the total variance of each estimator as a fraction of that of the standard estimator (that is, the ratio $\text{tr}(\text{Var}[\hat{\mathcal{G}}])/\text{tr}(\text{Var}[\hat{\mathcal{G}}_{n,m}])$) for Bayesian logistic regression with the `mushrooms` dataset. The ratios are between 60% and 70% for all methods, with the random subsets estimator showing the highest variance and the complete U-statistic the lowest. This confirms it is possible to reduce gradient variance with U-statistics. Moreover, the estimators can be ordered by their gradients' total variance. The complete U-statistic estimator and the 2nd order approximation have the smallest variance, the permuted-block estimator has slightly higher variance, and the random subsets estimator has the highest variance (but still less than that of the standard estimator). Recall that, according to Prop. 4.2, $\ell = 20$ implies that the permuted estimator achieves 95% of the variance reduction provided by the complete-U-statistic IW-ELBO. In this case, we estimated the variance reduction of the permuted-block estimator to be 91.24% of that of the complete-U-statistic estimator. We also show the ratio of the objective's variances in Figure 3–(b). Most estimators have a ratio of around 80%, but the permuted-block estimator achieves a 95.72% variance reduction provided by the complete U-statistic estimator.

Table 1: For Bayesian logistic regression (a) and Stan models (b), difference in nats of the average objective (higher is better) when trained using the permuted estimator vs. the standard IW-ELBO estimator. The variational distribution is a Gaussian distribution using either a full rank covariance matrix (first column) or a diagonal one (second column). The entry is "—" when the model diverged for all configurations, and NaN when it diverged for the specific configuration (other configurations are found in the Appendix).

(a) Bayesian logistic regression models.

| Dataset | permuted − standard IW-ELBO | |
| | $m = 8$ | |
| | Full Covariance | Diagonal |
| --- | --- | --- |
| a1a | 112.42 | 4.48 |
| australian | 3.36 | 1.38 |
| ionosphere | 16.58 | 0.06 |
| mushrooms | 202.56 | 8.69 |
| sonar | 50.62 | 0.19 |

(b) Stan models.

| Dataset | permuted − standard IW-ELBO | |
| | $m = 8$ | |
| | Full Covariance | Diagonal |
| --- | --- | --- |
| congress | 19.80 | 7.33 |
| election88 | 1133.70 | 6.94 |
| election88Exp | NaN | 32.76 |
| electric | 80.46 | 4.32 |
| electric-one-pred | -3.45 | -3.91 |
| hepatitis | NaN | 0.65 |
| hiv-chr | 283.19 | 15.84 |
| irt | 16077.03 | 1.00 |
| irt-multilevel | — | 62.32 |
| mesquite | 1.41 | 2.00 |
| radon | 268.98 | 14.83 |
| wells | -0.03 | -0.11 |

**VI Performance**    Ultimately, our goal is to provide a more efficient optimization method. To measure typical stochastic optimization performance, we first took the maximum objective value across learning rates in each iteration to construct the optimization *envelope* for each method and random seed [cf. Geffner & Domke (2018)]. The purpose of the envelope is to eliminate the learning rate as a nuisance parameter since stochastic optimization methods are very sensitive to learning rate, and one common benefit of variance reduction is to allow a larger learning rate. Then, for each method we used the median envelope across the 50 random seeds as a measure of its typical optimization behavior over iterations. Examples can be seen in Figure 2. As a final metric for each method we computed the *average objective* value (of the median envelope) across iterations up to 10,000 iterations,[5] excluding the first 50 iterations, which were highly noisy and sensitive to initialization. This is a useful summary metric to measure the tendency of one method to "stay ahead" of another (see the examples in Figure 2). Agrawal et al. (2020) found a similar metric effective for learning rate selection.

Table 2: Times for 1000 iterations of optimization with different estimators on the `mushrooms` dataset with $n = 24$, $m = 12$, averaged over 100 trials.

| Method | Time (s) | |
| | Mean | Std |
| --- | --- | --- |
| $\hat{\mathcal{L}}_{24,12}$ standard IW-ELBO | 5.47 | 0.04 |
| $\hat{\mathcal{L}}_{24,12}^{U}$ complete U | 1573.27 | 2.12 |
| $\hat{\mathcal{L}}_{\mathcal{S}_{20\frac{24}{12}}}$ random subsets | 6.49 | 0.09 |
| $\hat{\mathcal{L}}_{\mathcal{S}_{\Pi}}^{20}$ permuted block | 6.45 | 0.09 |
| $\hat{\mathcal{L}}_{24,12}^{\mathcal{A}}$ approx. | 5.25 | 0.02 |
| $\hat{\mathcal{L}}_{24,12}^{\mathcal{A},2}$ approx. 2nd order | 5.54 | 0.04 |

Table 1 shows the average objective difference between the permuted-block and standard IW-ELBO estimators for $m = 8$, with positive numbers indicating better performance for permuted-block. We focus on permuted-block here because it consistently achieves an excellent tradeoff between variance reduction and running time. In Appendix D we present similar results for two additional methods—the 2nd order approximation and the permuted-block estimator with DReG as the base gradient estimator—and for different values of $m$; in Appendix G we show the median envelopes themselves for many combinations of models, methods, and $m$. The examples in Figure 2 were selected to show cases where the difference is big (left), small (center), and negative (right); to contextualize our summary metric, we also added a reference vertical bar showing an iteration where the difference between the two envelopes is approximately

---

[5]For some datasets, such as `sonar`, we observed early convergence by visual inspection and computed the metric only up to that point. See Figures in Appendix G.

equal to the average objective difference. These results make it clear that the permuted-block estimator improves the convergence of stochastic optimization for VI across a range of models and settings. In `electric-one-pred`, permuted-block was consistently worse, but we verified that it still had lower-variance gradients; we speculate this is an unstable model where higher variance gradients help escape local optima.

**Running Time**  Table 2 shows the times required to complete 1000 iterations of optimization with different estimators for Bayesian logistic regression with the `mushrooms` data set, averaged over 100 trials. Here we used $n = 24$ and $m = 12$, which makes it a challenging setting for the complete U-statistic estimator, because there are $\binom{24}{12} = 2{,}704{,}156$ sets. As expected, the complete U-statistic is orders of magnitude slower. The approximations are faster than the standard estimator because the smallest $m - 1$ log-weights do not contribute to the objective, and thus their gradients are not needed. The permuted-block estimator incurs an extra cost of less than 1 ms per iteration compared to the standard IW-ELBO estimator for this model (a 18% increase). However, the increased time only depends on $m$, $n$, and $\ell$, and not on the model. Even for a very complex model, we would expect the extra time for these settings to be on the order of order of 1 ms per iteration, and be negligible compared to other costs. For example, for the `irt`, the standard estimator took 16.62s (0.11), while the permuted-block estimator took 17.54s (0.12), i.e., a 5% increase.

**Incomplete U-Statistics and Approximations**  Previously, we analyzed the methods by comparing them to the standard IW-ELBO estimator. In this part we will use the complete U-statistics as a baseline: given a realization of log-weights $v_1, \ldots, v_n$, we measure the difference between the objective value assigned by the complete U-statistic and the alternatives. For this experiment, we will use $n = 16$, $m = 8$ and the Bayesian logistic regression dataset `mushrooms`. In Figure 4 we plot the difference measured in nats as a function of the iteration step. From that plot (especially the inset), it is clear that the approximations are underestimators.

It is also interesting to see the approximations and the incomplete U-statistic being complementary: as the optimization progresses, the error of the approximations increases, but the error made by the incomplete U-statistics decreases. We expected this result because the variance of the log-weights decreases with the optimization. (The upper-bound of Eq. (10) is achieved when all $v_i$ are equal; but this is exactly the case when all the incomplete U-statistics coincide.)

**Dirichlet Experiments**  We conducted experiments with random Dirichlet distributions as described in (Domke & Sheldon, 2018). The goal was twofold. First, this is a setting where exact inference is possible, so we can evaluate IWVI with different estimators on the accuracy of posterior inference directly, instead of using the IW-ELBO as a proxy. Secondly, this is a simple setting to demonstrate that the optimal value of $m$ is often strictly between 1 and $n$, which is the regime in which our variance reduction methods are useful (all but the approximations coincide when $m \in \{1, n\}$). We again used SGD with 15 different learning rates and selected,

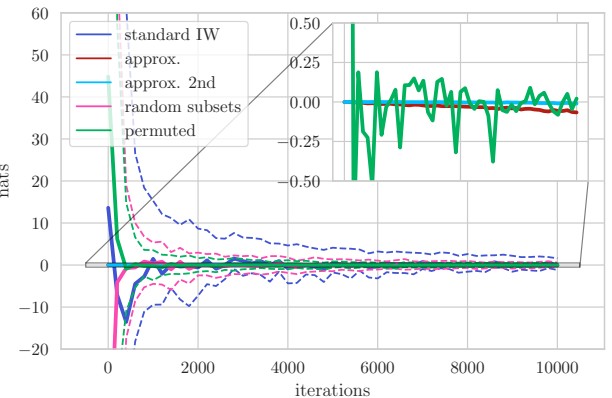

Figure 4: Difference between estimated value using any of the methods and that of the complete U-statistic, in nats, for the `mushrooms` dataset. (25th and 75th percentiles shown with dashed lines.) As optimization progresses, the error of the incomplete U-statistics decreases, but the error of the approximation increases. The inset shows the permuted and both approximations in a region that is 0.5 nats of the target value.

for each configuration, the learning rate that achieved the best mean objective after 10k iterations. We optimized each configuration using, for this experiment, 100 different random seeds. We estimated the accuracy of the approximation by computing the distance (error) between the distribution's covariance and the estimated covariance of the learned approximation. Figure 1 shows the error as a function of $m$ for different values of $n$ when using the standard IW-ELBO estimator for a random Dirichlet with 50 parameters. The figure shows that the optimal $m$ increases with $n$, but slowly. Figure 5 shows similar results for other esti-

mators: permuted, DReG, and permuted-DReG. In all cases, we confirm that, for this model, the optimal $m$ lies strictly between 1 and $n$. We provide in Appendix E additional details.

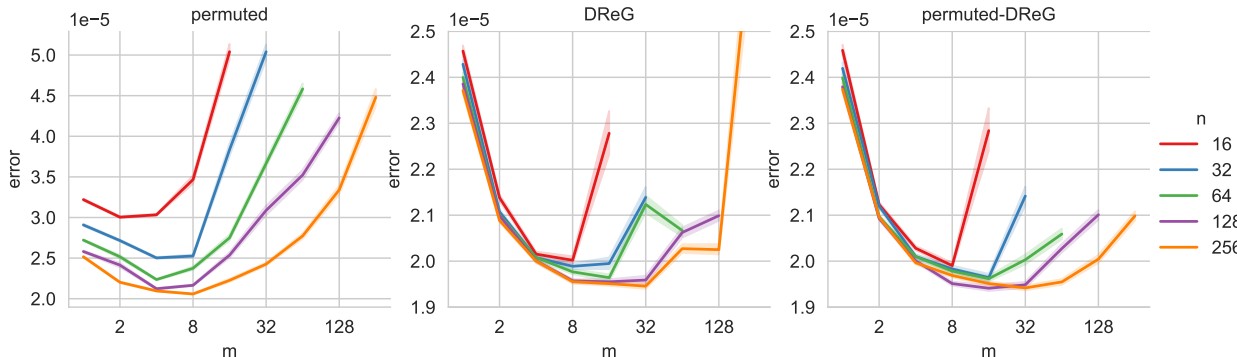

Figure 5: Distance between the covariance of a random Dirichlet distribution with 50 parameters and the covariance of its approximation as a function of $m$ for different values of $n$ after training using the permuted (left), DReG (center) or permuted-DReG (right) estimators.

## 6.1 Importance-Weighted Autoencoders

To evaluate the performance of the proposed methods on IWAEs, we trained IWAEs on 4 different datasets: `MNIST`, `KMNIST`, `FMNIST`, and `Omniglot`. We compare the standard IW-ELBO estimator and DReG estimators to their permuted versions, i.e., the permuted and permuted-DReG estimators. We also evaluate the second-order approximation to the complete-U-statistic estimator. We trained each combination of dataset, method, and value of $m$ using five different random seeds, and the optimization was run for 100 epochs using Adam (Kingma & Ba, 2015).

In Figure 6, we present the final testing objective for different values of $m$ (using $n = 50$ in all cases) for the `KMNIST` dataset, and we show results for the rest of the datasets in Figure 8 in the Appendix F along with further details on the experiments. The figure shows that the permuted versions consistently improved over the base versions, i.e., the permuted estimator improves over the standard-IW estimator in the same way as the permuted-DReG estimator improves over the DReG estimator. Additionally, we can see that the second-order approximation outperforms the permuted estimator for small values of $m$. However, as $m$ increases, the permuted estimator takes the lead, which is expected since the approximation error grows with $m$.

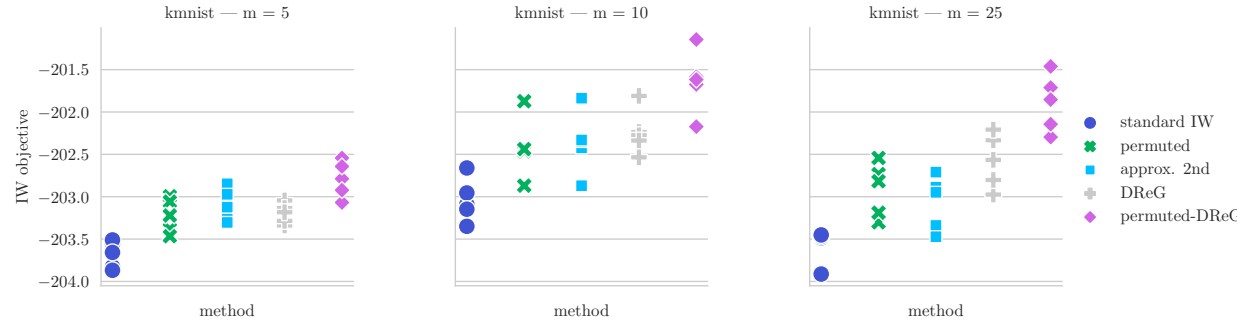

Figure 6: Objective's distribution for `KMNIST` with $n = 50$ and different combinations of methods and $m$.

We also compared the total wall-clock time required to complete the optimization with different estimators in Figure 9 in the Appendix. It can be seen that there is not a significant time increase for using our proposed methods.

## 7 Related and Future Work

Gradient variance reduction is an active topic in VI because of its impact on stochastic optimization. Our complete- and incomplete-U-statistic methods are complementary to other variance reduction techniques: they are compatible with different base estimators, including the Doubly Reparameterized Gradient Estimator (DReG) of Tucker et al. (2018) and the generalization of Bauer & Mnih (2021). Another broad approach to variance reduction is the use of control variates (Miller et al., 2017; Mnih & Gregor, 2014; Ranganath et al., 2014; Geffner & Domke, 2018; 2020). In the case of IWVI, the control variates of Mnih & Rezende (2016) and Liévin et al. (2020), which are designed for the score function estimator, could work as a base estimator from which a U-statistic can be built. We leave its empirical evaluation for future work.

Importance-weighted estimators are also being used for the Reweighted Wake-Sleep (RWS) procedure (Bornschein & Bengio, 2015; Le et al., 2020) and its variations (Dieng & Paisley, 2019; Kim et al., 2020). Given the connection between the gradient estimators of RWS and that of the IW-ELBO [see Kim et al. (2020)], these estimators could be potentially improved by using the ideas of the complete- and incomplete-U-statistic methods.

The numerical approximations of Section 5 follow a different principle of approximating the objective; it is an open question if such an approximation can be used in conjunction with other variance reduction methods. Interestingly, the first-order approximation expresses the objective as a convex combination of the ordered log-weights (minus a constant), which has a form similar to the objective presented in Wang et al. (2018), albeit with different coefficients.

It would be an interesting future line of work to extend the order of Proposition 4.1 to a partial order of random variables in the sense of Mattei & Frellsen (2022).

Nowozin (2018) introduced Jackknife-VI (JVI), which uses complete-U statistics to reduce bias instead of variance. In Appendix A we briefly discuss possible applications of our methods to JVI.

## 8 Conclusion

We introduced novel methods based on U-statistics to reduce gradient and objective variance for importance-weighted variational inference, and found empirically that the methods improve black-box VI performance and IWAEs training. We recommend using the permuted-block estimator in any situation with $r > 1$ replicates: it never increases variance, and can be tuned based on computational budget to achieve any desired fraction of the possible variance reduction. In practice, a 95% fraction of possible variance reduction can be achieved at a very low cost. The approximations of Section 5 are extremely fast and provide substantial variance reduction, but are not universally better than the standard estimator because they introduce some bias that can hurt performance, especially in easier models near the end of optimization.

**Acknowledgments**

This material is based upon work supported by the National Science Foundation under Grant Nos. 1749854 and 1908577. JB would like to thank Tomás Geffner and Miguel Fuentes for their helpful discussions.

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

## A   Experiments with Jackknife

The relation between the jackknife estimator and complete U-statistics was made explicit early on by Mantel (1967). Recently, Nowozin (2018) used the jackknife estimator as a way to diminish the bias in IW-VI, proposing jackknife VI (JVI). Using the notation of Section 3, the jackknife estimator is

$$\hat{\mathcal{L}}_n^{J,r}(V_{1:n}) = \sum_{j=0}^{r} c(n,r,j)\hat{\mathcal{L}}_{n,n-j}^{U}(V_{1:n}), \tag{14}$$

where $\hat{\mathcal{L}}_{n,n-j}^{U}$ is the complete U-statistic IW-ELBO estimator, and the $c(n,r,j)$ are the Sharot coefficients [cf. Nowozin (2018)].

In the original version (14), it evaluates a collection of $r$ complete U-statistics with $m$ ranging from $n$ to $n-r$. However, there is no need to constrain $m$ in that way, i.e., we can instead compute the following estimator

$$\hat{\mathcal{L}}_{n,m}^{J,r}(V_{1:n}) = \sum_{j=0}^{r} c(m,r,j)\hat{\mathcal{L}}_{n,m-j}^{U}(V_{1:n}), \qquad \text{for } r < m \leq n,$$

because the bias is a function of $m$. This means that once $m$ is fixed, we can pick the number of independent samples $n \geq m$ to reduce the variance of the estimation.

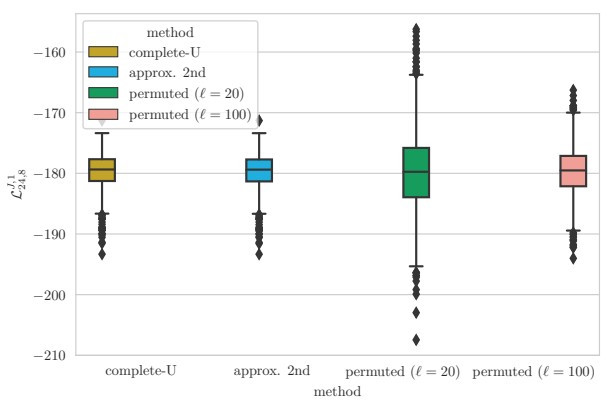

Figure 7: Distributions of the objective using the Jackknife estimator $\hat{\mathcal{L}}_{24,8}^{J,1}$ on an approximation of the posterior of the mushrooms dataset, using different estimators.

of building the index set.[7]

For our experiment, we optimized a variational approximation to the posterior of the mushrooms dataset as in Section 6. We used the complete-U-statistic IW-ELBO estimator for optimization ($n' = 16$ and $m = 8$), and we choose the configuration with the highest final bound.

We evaluated the trained model using the Jackknife estimator with $n = 24$, $r = 1$ and $m = 8$. For the inner estimator we used the complete-U-statistic IW-ELBO estimator, a variation of the permuted-block IW-ELBO estimator[6] with $\ell = 20$ and with $\ell = 100$, and the second order approximation. Figure 7 shows that, when using the permuted estimator with $\ell = 20$, the increased variance gets translated into an increased variance in the final estimation. However, it can be reduced by increasing the number of permutations to $\ell = 100$. In the following table, we show the time taken to compute the Jackknife estimator without accounting for the time

| Method | Mean time (ms) | Std |
|---|---:|---|
| complete-U | 23.98 | 2.28 |
| approx. 2nd | 0.71 | 0.05 |
| permuted ($\ell = 20$) | 0.69 | 0.02 |
| permuted ($\ell = 100$) | 0.86 | 0.03 |

In this case, we observed that the alternatives are approximately 30 times faster than using the complete-U-statistic.

---

[6]Since $n$ is not an integer multiple of $m - 1$, we reduced the total number of sets to $\ell m \lfloor \frac{n}{m} \rfloor$.

[7]We pre-computed the index set for the complete-U-statistic, which in this case requires $735471 + 346104 = 1081575$ sets, taking 1.34 seconds.

# B    Additional Theoretical results

In this section, we apply a result of Halmos (1946) to the estimation of the IW-ELBO. Subject to certain conditions, the estimator $\hat{\mathcal{L}}_{n,m}^U$ has the smallest variance of any unbiased estimator of the IW-ELBO. The technical conditions are needed to define the class of "unbiased estimators" as ones that are unbiased for all log-weight distributions in a non-trivial class.

**Proposition B.1.** *Let $\mathbb{E}_F[\cdot]$ and $\mathrm{Var}_F[\cdot]$ denote expectation and variance with respect to log-weights $V_1, \ldots, V_n$ drawn independently from distribution $F$, and let $\mathcal{L}_m(F) = \mathbb{E}_F\left[\ln \frac{1}{m}\left(\sum_{i=1}^m e^{V_i}\right)\right]$ be the IW-ELBO with log-weight distribution $F$. Let $\tilde{\mathcal{F}}$ denote the set of distributions supported on a finite subset of $\mathbb{R}$. Suppose $\Phi$ is any estimator such that $\mathbb{E}_{\tilde{F}}[\Phi(V_{1:n})] = \mathcal{L}_m(\tilde{F})$ for all $\tilde{F} \in \tilde{\mathcal{F}}$. Then,*

$$\mathrm{Var}_F[\hat{\mathcal{L}}_{n,m}^U(V_{1:n})] \leq \mathrm{Var}_F[\Phi(V_{1:n})]$$

*whenever the latter quantity is defined, for* any *distribution $F$ on the real numbers (up to conditions of measurability and integrability).*

*Proof.* The result is a direct application of Theorem 5 of Halmos (1946). $\qquad\square$

For IW-ELBO estimation, the conditions are rather mild: we expect an IW-ELBO estimator to work for generic log-weight distributions. For gradient estimation, we take the conclusion lightly, because gradient estimators often use specific properties of the underlying distributions, such as having a reparameterization.

## B.1    Additional Proofs

In this section, we provide a proof of Proposition 3.3. We first need to define a quantity similar to Definition 3.1. Recall, from the statement of the Proposition, that $g \colon \mathbb{R}^{d_Z} \to \mathbb{R}^{d_\phi}$, and let $g_i$ denote its $i$th component.

**Definition B.2.** *Let $Z_1, \ldots, Z_{2m}$ be i.i.d. drawn from $q_\phi$, and $1 \leq i \leq d_\phi$. For $0 \leq c \leq m$, take $\mathbf{s}, \mathbf{s}' \in \binom{[\![2m]\!]}{m}$ with $|\mathbf{s} \cap \mathbf{s}'| = c$. Using g from* Proposition 3.3, *define*

$$\varsigma_c^{(i)} = \mathrm{Cov}\Big[g_i(Z_{s_1}, \ldots, Z_{s_m}),\, g_i(Z_{s_1'}, \ldots, Z_{s_m'})\Big],$$

*which depends only on c and not the particular $\mathbf{s}$ and $\mathbf{s}'$.*

We can now proceed with the proof.

*Proof of* Proposition 3.3. For $1 \leq i \leq d_\phi$, using Eq. (4) and (6), it follows from Theorem 5.2 of Hoeffding (1948) that
$$\mathrm{Var}[(\hat{\mathcal{G}}_{n,m}^U)_i] \leq \tfrac{m}{n}\varsigma_m^{(i)} = \mathrm{Var}[(\hat{\mathcal{G}}_{n,m})_i].$$
From the definition of the covariance matrix, we get

$$\mathrm{tr}(\mathrm{Var}[\hat{\mathcal{G}}_{n,m}^U]) = \sum_{i=1}^{d_\theta} \mathrm{Var}[(\hat{\mathcal{G}}_{n,m}^U)_i] \leq \sum_{i=1}^{d_\theta} \mathrm{Var}[(\hat{\mathcal{G}}_{n,m})_i] = \mathrm{tr}(\mathrm{Var}[\hat{\mathcal{G}}_{n,m}]).$$

Using again that $\hat{\mathcal{G}}_{n,m}^U$ and $\hat{\mathcal{G}}_{n,m}$ are unbiased, that is, $\mathbb{E}[\hat{\mathcal{G}}_{n,m}^U] = \mathbb{E}[\hat{\mathcal{G}}_{n,m}]$, then

$$\mathbb{E}\,\|\hat{\mathcal{G}}_{n,m}^U\|_2^2 = \sum_{i=1}^{d_\theta}(\mathrm{Var}[(\hat{\mathcal{G}}_{n,m}^U)_i] + \mathbb{E}[(\hat{\mathcal{G}}_{n,m}^U)_i]^2) \leq \sum_{i=1}^{d_\theta}(\mathrm{Var}[(\hat{\mathcal{G}}_{n,m})_i] + \mathbb{E}[(\hat{\mathcal{G}}_{n,m})_i]^2) = \mathbb{E}\,\|\hat{\mathcal{G}}_{n,m}\|_2^2.$$

$\square$

## C Dataset description

We provide a brief description of the datasets and models used for the experiments. The models used for Bayesian logistic regressions were taken from the UCI Machine Learning Repository Dua & Graff (2017). The rest of the models are part of the Stan Example models Stan Development Team (2021); Carpenter et al. (2017).

For the dataset used for Bayesian logistic regression, whenever there was a categorical variable with $k$ categories, we *dummified* it by creating $k-1$ dummies variables. Additionally, for the `a1a` dataset, continuous variables were discretized into quintiles following the work of Platt (1999). However, since we were unable to find the file describing the actual process used for the discretization, some discrepancies remained.

Table 3: Description of datasets/models.

| Name | Num. of variables | Num. of records | Comments |
|---|---|---|---|
| a1a | 105 | 1605 | First 1605 instances of the Adult Data Set, following LIBSVM Chang & Lin (2011), + discretized continuos and dummified. |
| australian | 35 | 690 | From UCI + dummified. |
| ionosphere | 35 | 351 | From UCI |
| mushrooms | 96 | 8124 | From UCI + dummified. |
| sonar | 61 | 208 | From UCI |
| congress | 4 | 343 | Gelman & Hill (2006) Ch. 7 |
| election88 | 95 | 2015 | Gelman & Hill (2006) Ch. 19 |
| election88Exp | 96 | 2015 | Gelman & Hill (2006) Ch. 19 |
| electric | 100 | 192 | Gelman & Hill (2006) Ch. 23 |
| electric-one-pred | 3 | 192 | Gelman & Hill (2006) Ch. 23 |
| hepatitis | 218 | 288 | WinBUGS Lunn et al. (2000) examples |
| hiv-chr | 173 | 369 | Gelman & Hill (2006) Ch. 7 |
| irt | 501 | 30105 | Gelman & Hill (2006) Ch. 14 |
| irt-multilevel | 604 | 30015 | Gelman & Hill (2006) Ch. 14 |
| mesquite | 3 | 46 | Gelman & Hill (2006) Ch. 4 |
| radon | 88 | 919 | `radon-chr` from Gelman & Hill (2006) Ch. 19 |
| wells | 2 | 3020 | Gelman & Hill (2006) Ch. 7 |
| MNIST | 784 | 60000 + 10000 | LeCun et al. (2010) |
| FMNIST | 784 | 60000 + 10000 | Fashion-MNIST, Xiao et al. (2017) |
| KMNIST | 784 | 60000 + 10000 | Kuzushiji-MNIST Clanuwat et al. (2018) |
| Omniglot | 784 | 24345 + 8070 | Lake et al. (2015) from Burda et al. (2016) |

## D Pairwise comparison

In this section we present the mean difference of the medians of the envelopes as described in Section 6. We compare the methods that used the reparameterized gradients as based gradient estimator, i.e., the permuted-block estimator and the 2nd order approximation, to the standard IW-ELBO estimator. Additionally, we compare the standard IW using DReG as a based gradient estimator with a version of the permuted-block that uses the DReG as a base gradient estimator, namely, the permuted DReG.

Interestingly, in the settings presented in Table 7, only the proposed methods, i.e., the complete-U statistic with its two approximations, the permuted-block, and the random subsets, converged at some point. All the other methods diverged, which explains why we cannot compute the difference.

Table 4: Bayesian logistic regression models using a Gaussian approximation with a covariance matrix of full rank. Difference in nats of the average objective (higher values are better).

| Dataset | permuted - standard IW | | | approx. 2nd - standard IW | | | permuted DReG - DReG | | |
|---|---|---|---|---|---|---|---|---|---|
| | $m$ | | | | | | | | |
| | 2 | 4 | 8 | 2 | 4 | 8 | 2 | 4 | 8 |
| a1a | 45.36 | 100.56 | 112.42 | 47.53 | 105.30 | 122.34 | 27.30 | 111.74 | 119.51 |
| australian | 1.31 | 2.61 | 3.36 | 1.07 | 2.37 | 3.22 | 1.45 | 1.87 | 3.94 |
| ionosphere | 3.89 | 13.17 | 16.58 | 4.11 | 13.55 | 17.55 | 4.34 | 15.74 | 17.91 |
| mushrooms | 64.46 | 145.85 | 202.56 | 67.28 | 153.58 | 214.31 | 93.45 | 186.01 | 179.02 |
| sonar | 30.15 | 61.09 | 50.62 | 32.94 | 63.34 | 54.14 | 27.99 | 69.54 | 90.86 |

Table 5: Bayesian logistic regression models using a diagonal Gaussian approximation. Difference in nats of the average objective (higher values are better).

| Dataset | permuted - standard IW | | | approx. 2nd - standard IW | | | permuted DReG - DReG | | |
|---|---|---|---|---|---|---|---|---|---|
| | $m$ | | | | | | | | |
| | 2 | 4 | 8 | 2 | 4 | 8 | 2 | 4 | 8 |
| a1a | 1.54 | 4.01 | 4.48 | 1.49 | 4.08 | 4.45 | 1.40 | 12.67 | 12.86 |
| australian | 0.02 | 1.00 | 1.38 | 0.05 | 0.96 | 1.28 | -0.07 | 0.06 | 1.43 |
| ionosphere | -0.10 | -0.10 | 0.06 | -0.12 | -0.21 | 0.00 | -0.12 | -0.08 | 0.31 |
| mushrooms | 1.88 | 2.76 | 8.69 | 1.74 | 3.30 | 9.16 | 1.94 | 4.69 | 8.50 |
| sonar | 0.03 | -0.15 | 0.19 | -0.02 | -0.18 | 0.15 | 0.03 | -0.28 | 0.21 |

Table 6: Stan models using a diagonal Gaussian approximation. Difference in nats of the average objective (higher values are better).

| Dataset | permuted - standard IW | | | approx. 2nd - standard IW | | | permuted DReG - DReG | | |
|---|---|---|---|---|---|---|---|---|---|
| | $m$ | | | | | | | | |
| | 2 | 4 | 8 | 2 | 4 | 8 | 2 | 4 | 8 |
| congress | 2.50 | 4.61 | 7.33 | 2.89 | 4.76 | 7.63 | 2.37 | 4.68 | 7.02 |
| election88 | 0.12 | 2.66 | 6.94 | 0.12 | 2.84 | 7.06 | 0.10 | 2.67 | 6.83 |
| election88Exp | 0.82 | 98.52 | 32.76 | 4.73 | 117.78 | 55.27 | -1.89 | 100.09 | 32.53 |
| electric | 0.26 | 1.53 | 4.32 | 0.16 | 1.54 | 4.52 | 0.24 | 1.56 | 4.63 |
| electric-one-pred | 0.66 | -0.77 | -3.91 | 0.74 | -0.76 | -4.38 | 0.69 | -0.77 | -3.93 |
| hepatitis | 0.90 | -0.06 | 0.65 | 2.06 | 156.53 | 1.86 | -0.30 | 0.92 | 0.69 |
| hiv-chr | 0.16 | 2.03 | 15.84 | 0.34 | 2.12 | 21.74 | -0.08 | 1.45 | 12.91 |
| irt | 0.19 | 0.80 | 1.00 | 0.15 | 0.72 | 0.93 | 0.11 | 0.61 | 1.40 |
| irt-multilevel | 35.69 | 43.79 | 62.32 | 29.74 | 48.20 | 53.64 | 34.66 | 50.26 | 55.22 |
| mesquite | 0.20 | 0.58 | 2.00 | -0.06 | 0.28 | 1.74 | -0.29 | 0.39 | 1.99 |
| radon | 7.88 | 5.79 | 14.83 | 7.85 | 8.91 | 65.49 | 8.16 | 7.56 | 60.92 |
| wells | -0.02 | 0.01 | -0.11 | -0.20 | -0.30 | -0.35 | -0.02 | -0.04 | -0.14 |

Table 7: Stan models using a full covariance Gaussian approximation. Difference in nats of the average objective (higher values are better).

| Dataset | permuted - standard IW | | | approx. 2nd - standard IW | | | permuted DReG - DReG | | |
|---|---|---|---|---|---|---|---|---|---|
| | $m$ | | | | | | | | |
| | 2 | 4 | 8 | 2 | 4 | 8 | 2 | 4 | 8 |
| congress | 11.62 | 12.02 | 19.80 | 12.33 | 12.57 | 20.46 | 13.55 | 13.11 | 20.96 |
| election88 | NaN | 1785 | 1133 | NaN | 2494 | 2170 | NaN | 1776 | 1116 |
| election88Exp | NaN | NaN | NaN | NaN | NaN | NaN | NaN | NaN | NaN |
| electric | NaN | -38.02 | 80.46 | NaN | -77.53 | 89.06 | NaN | -43.16 | 34.91 |
| electric-one-pred | -1.81 | -4.73 | -3.45 | -3.18 | -4.77 | -4.37 | -1.79 | -4.72 | -3.46 |
| hepatitis | NaN | NaN | NaN | NaN | NaN | NaN | NaN | NaN | NaN |
| hiv-chr | NaN | NaN | 283.19 | NaN | NaN | 325.79 | NaN | NaN | NaN |
| irt | 17793 | 20064 | 16077 | 19399 | 22000 | 17686 | NaN | NaN | NaN |
| mesquite | 2.57 | 1.20 | 1.41 | 2.43 | 0.95 | 1.19 | 2.67 | 0.53 | 0.74 |
| radon | NaN | 1150 | 268.98 | NaN | 1316 | 303.83 | NaN | 11675 | 269.26 |
| wells | 0.02 | 0.07 | -0.03 | -0.29 | -0.31 | -0.29 | 0.04 | 0.02 | -0.02 |

# E  Random Dirichlet experiment

We follow Domke & Sheldon (2018) for the Random Dirichlet experiment. For a randomly-sampled Dirichlet Distribution with 50 parameters, we approximate it using a $(50 - 1)$-dimensional Gaussian distribution parameterized with a full rank covariance matrix, with its domain constrained to the simplex using PyTorch's distributions (Paszke et al., 2019).

We optimize each configuration using 100 different random seeds. We select the learning rate that achieved the highest objective among all learning rates that converged for all seeds. For each seed, we compute the Frobenius norm between the empirical covariance of the approximating distribution and that of the theoretical distribution (the error). The distribution of this error is shown in Figure 1 and 5. We had to exclude eight outliers with errors greater than $10^{-4}$ and up to 0.5. Interestingly, those outliers used either the DReG or permuted-DReG estimators.

# F  VAE details.

For the variational autoencoders, we used, for all datasets, the architecture used by Burda et al. (2016). We trained each configuration for a fixed number of epochs (100) using Adam (Kingma & Ba, 2015) with a learning rate of $10^{-4}$. In all cases, we used a batch size of 500, and a latent variable of dimension 50, while taking $n = 50$ samples. Datasets were taken from PyTorch, except for the `Omniglot`, for which we used the construction provided by Burda et al. (2016). We evaluated using the standard IW-ELBO estimator, regardless of the estimator used for the optimization.

To get consistent wall-clock time measurements, we trained only using CPU on dedicated servers, with disabled hyper-threading and a single task per core. Additionally, we used `set_flush_denormal` to avoid creating denormal numbers because some estimators create many of such numbers (especially DReG-like estimators), having a substantial negative impact on performance. Our implementation of DReG is based on Pyro's (Bingham et al., 2018) not-yet-integrated implementation. We are not aware of a PyTorch implementation without the extra time penalty.

In the following plots, we provide the objective distribution for all dataset/method/$m$ configurations and the distribution of the wall-clock time.

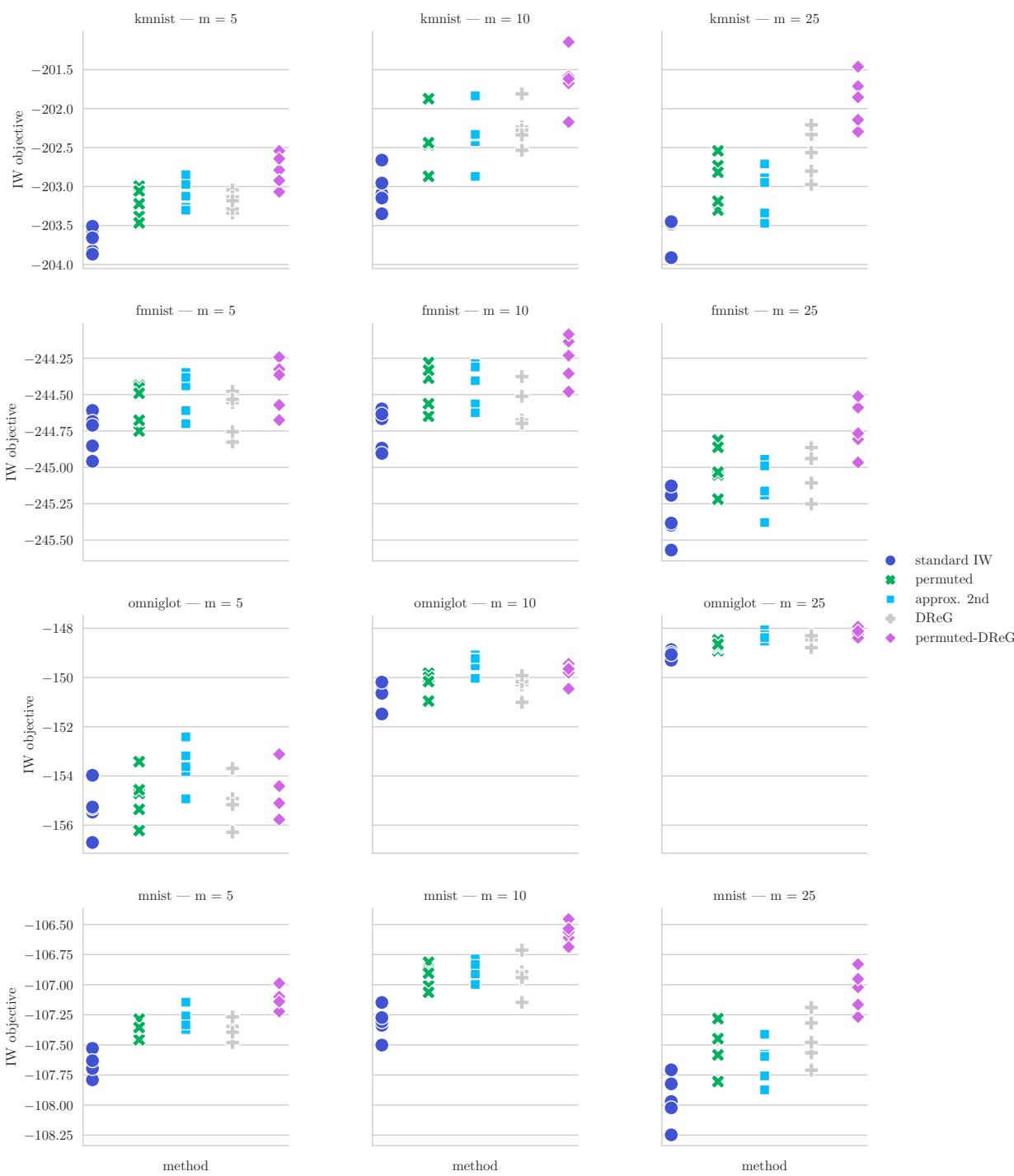

Figure 8: Distribution of the objective for different combinations of datasets, methods and $m$, using $n = 50$ samples.

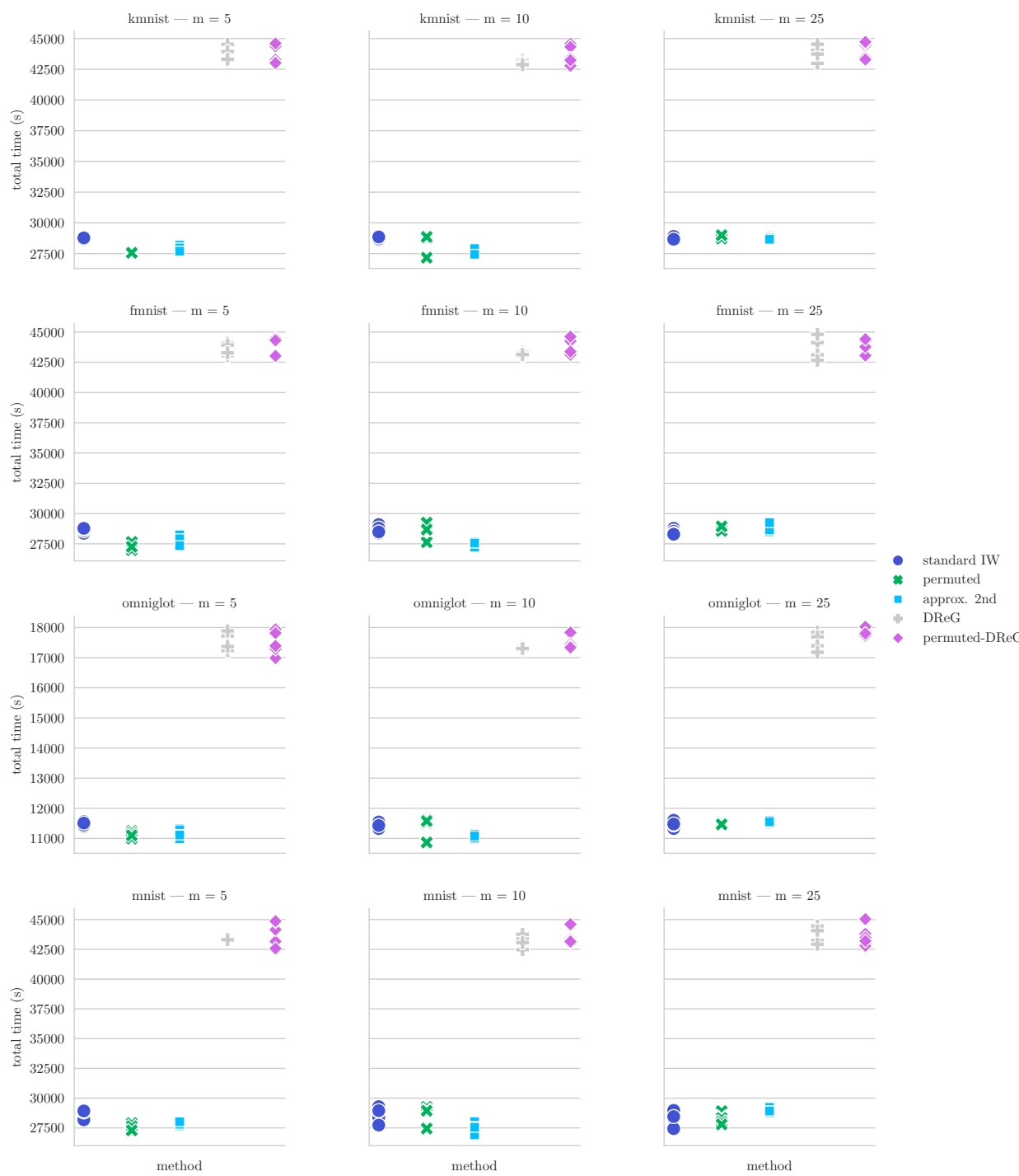

Figure 9: Distribution of the time taken to run 100 epochs for different combinations of datasets, methods and $m$, using $n = 50$ samples.

# G   Figures of median envelope

For some of the methods presented in the paper, we compute the median envelope during training as described in Section 6.

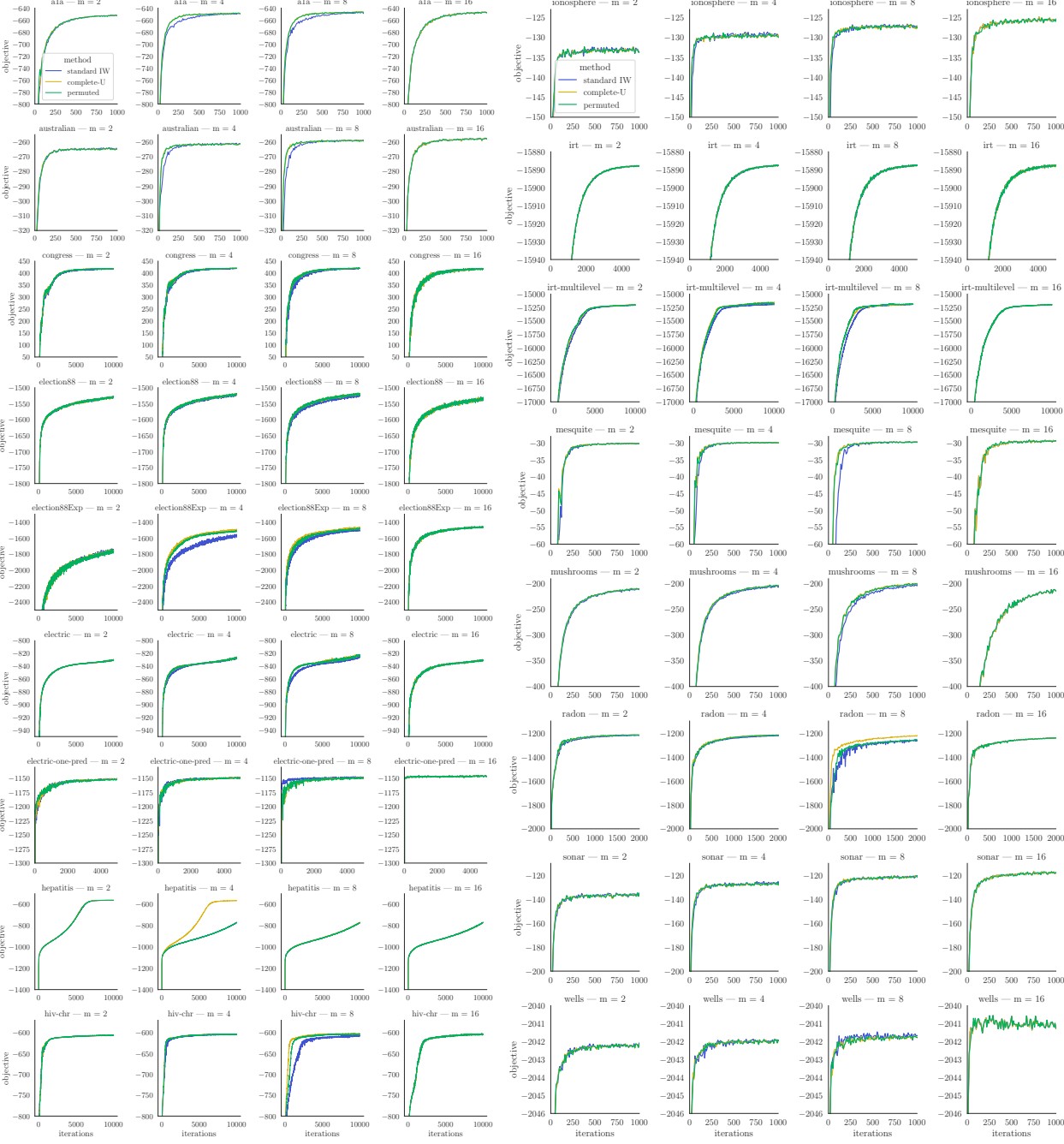

Figure 10: Median envelope for models when using a diagonal Gaussian as approximating distribution for the estimators complete-U, permuted and the standard IW.

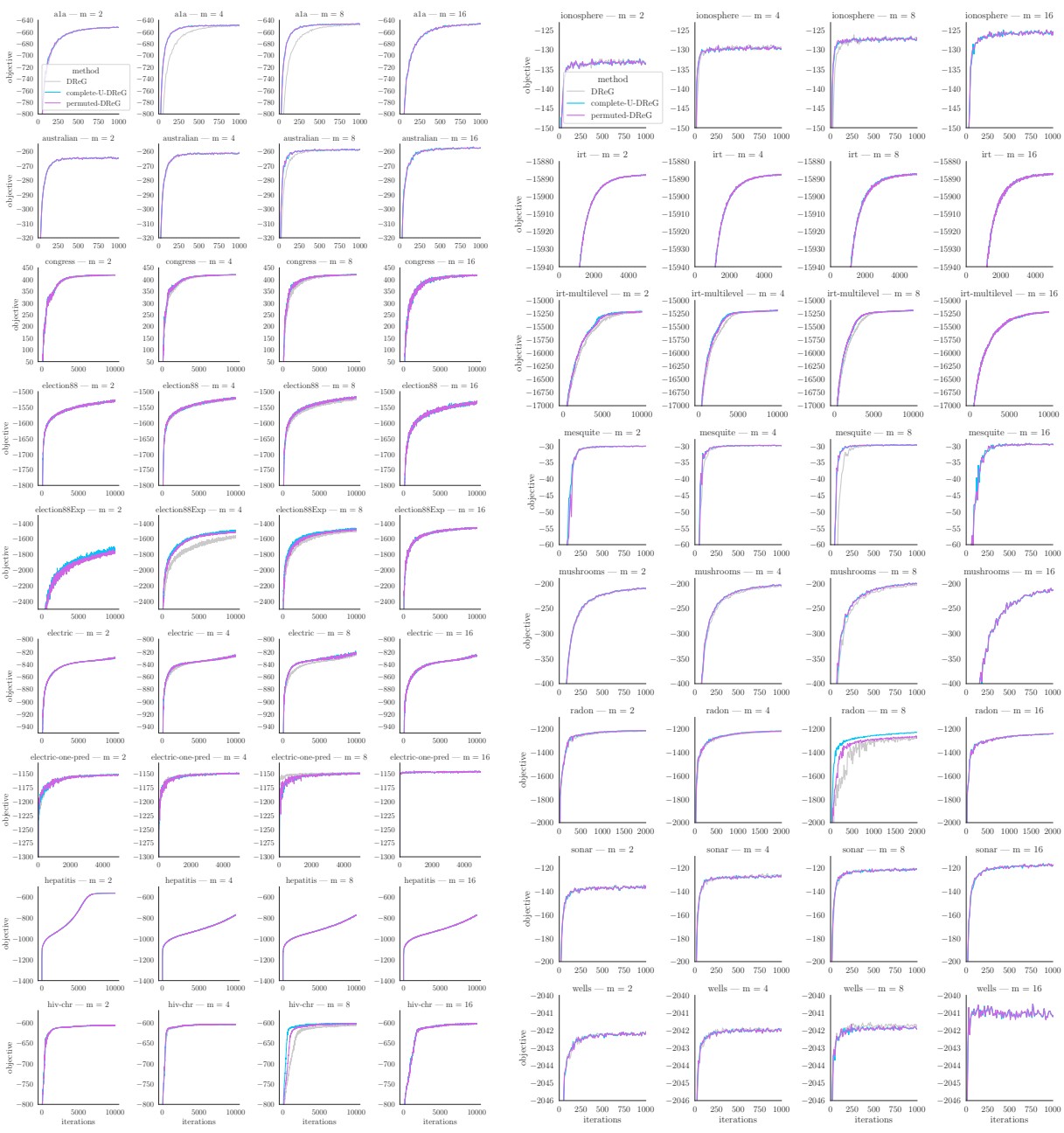

Figure 11: Median envelope for models when using a diagonal Gaussian as approximating distribution using the complete-U DReG, permuted DReG and the standard DReG gradient estimators.

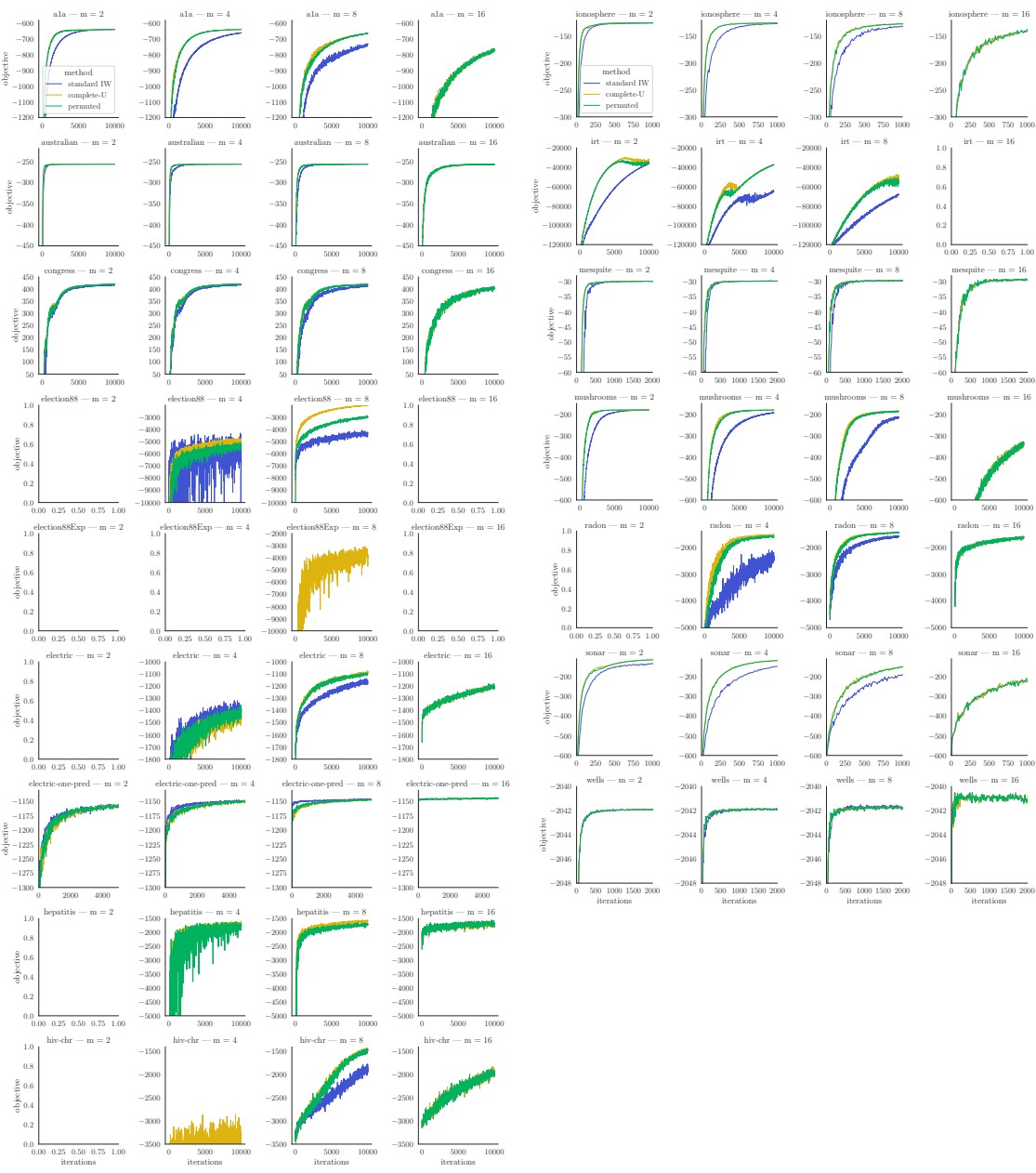

Figure 12: Median envelope for models when using a Gaussian distribution with full-rank covariance as approximating distribution for the estimators complete-U, permuted and the standard IW.

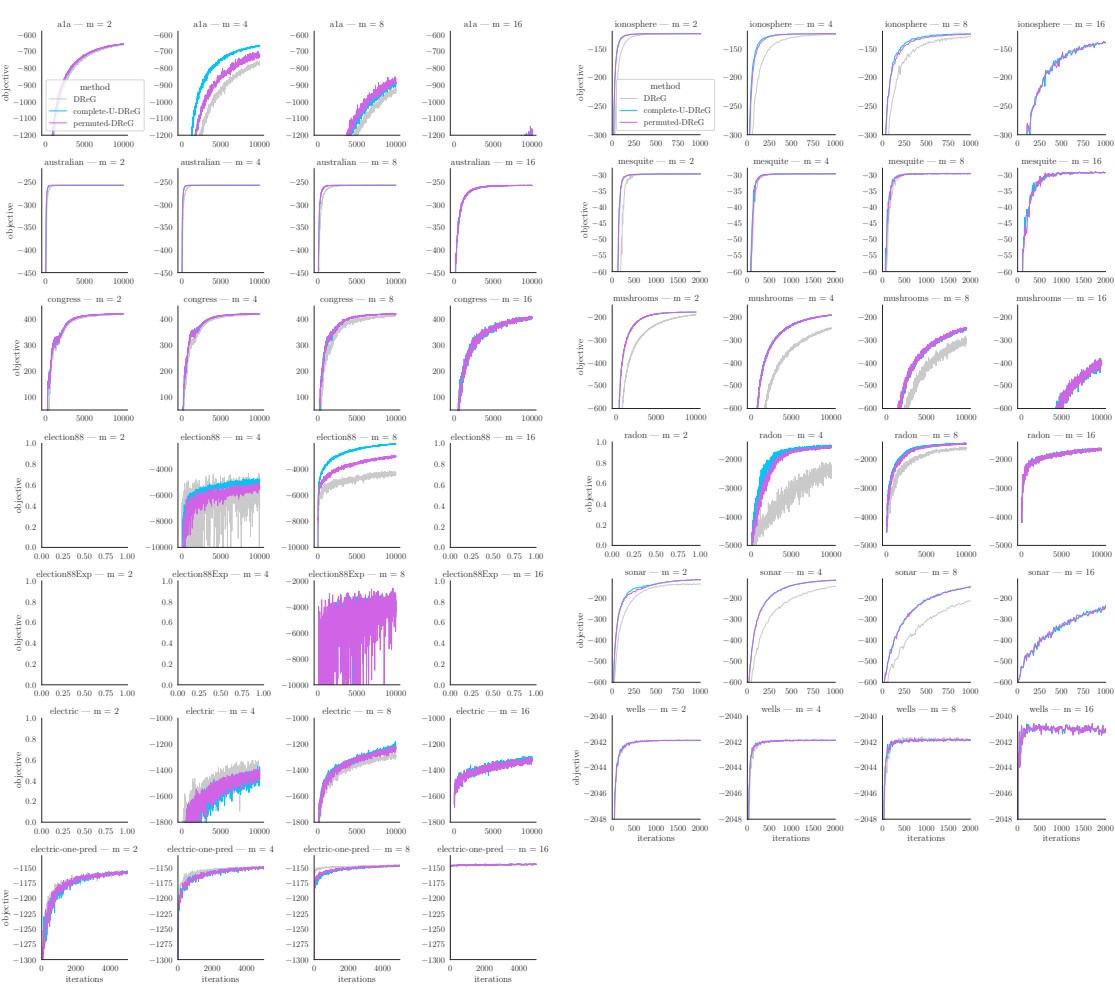

Figure 13: Median envelope for models when using a Gaussian distribution with full-rank covariance as approximating distribution using the complete-U DReG, permuted DReG and the standard DReG gradient estimators.

Table 8: Median objective averaged over the last 200 iterations when using the estimators complete-U, permuted and the standard IW. It can be seen that for at least 10 models out of 17, using either the diagonal Gaussian or the full rank covariance Gaussian approximation, the best objective is achieved with an intermediate value of $m$, and it is at least 1 nat larger than the objective with $m = 16$. These models are: `congress`, `election88`, `election88Exp`, `electric`, `electric-one-pred`, `hepatitis`, `hiv-chr`, `irt-multilevel`, `mushrooms` and `radon`. Optimizations using $m = 1$ are not shown.

| model | method | Diagonal Gaussian | | | | Full Rank Covariance Gaussian | | | |
|---|---|---|---|---|---|---|---|---|---|
| | | | | | m | | | | |
| | | 2 | 4 | 8 | 16 | 2 | 4 | 8 | 16 |
| a1a | standard IW | -652.8 | -649.7 | -648.0 | -647.6 | -639.0 | -660.7 | -738.1 | -772.1 |
| | complete-U | -652.6 | -648.9 | **-646.7** | -647.6 | **-637.8** | -639.0 | -663.9 | -772.1 |
| | permuted | -652.6 | -649.0 | -647.0 | -647.7 | -637.8 | -639.2 | -664.1 | -771.2 |
| australian | standard IW | -264.4 | -261.2 | -259.1 | -258.0 | **-256.8** | -256.8 | -256.8 | -257.2 |
| | complete-U | -264.8 | -261.5 | -259.1 | -258.0 | -256.8 | -256.8 | -256.8 | -257.2 |
| | permuted | -264.7 | -261.4 | -259.0 | **-257.9** | -256.8 | -256.8 | -256.8 | -257.1 |
| congress | standard IW | 417.1 | 419.5 | 419.5 | 417.6 | 416.9 | 417.2 | 412.4 | 403.9 |
| | complete-U | 418.8 | 420.3 | 420.3 | 417.6 | 419.4 | **420.3** | 419.2 | 403.9 |
| | permuted | 418.5 | 420.2 | **420.4** | 417.1 | 419.4 | 420.2 | 418.9 | 402.9 |
| election88 | standard IW | -1529.5 | -1523.3 | -1525.0 | -1535.6 | NaN | -5964.2 | -4383.2 | NaN |
| | complete-U | -1529.7 | -1521.5 | **-1519.8** | -1535.6 | NaN | -4943.1 | **-2046.2** | NaN |
| | permuted | -1529.4 | -1520.6 | -1520.5 | -1535.5 | NaN | -5443.7 | -3000.9 | NaN |
| election88Exp | standard IW | -1755.7 | -1570.8 | -1502.7 | **-1461.2** | NaN | NaN | NaN | NaN |
| | complete-U | -1760.0 | -1496.5 | -1467.0 | -1461.2 | NaN | NaN | **-3748.2** | NaN |
| | permuted | -1766.6 | -1512.3 | -1482.7 | -1461.2 | NaN | NaN | NaN | NaN |
| electric | standard IW | -830.5 | -827.7 | -826.1 | -830.9 | NaN | -1421.1 | -1166.2 | -1207.2 |
| | complete-U | -830.6 | -827.0 | **-823.0** | -830.9 | NaN | -1459.2 | **-1090.3** | -1207.2 |
| | permuted | -830.6 | -827.1 | -823.5 | -830.9 | NaN | -1413.6 | -1098.1 | -1203.4 |
| electric-one-pred | standard IW | -1148.6 | -1147.5 | -1146.4 | **-1144.6** | -1153.0 | -1145.8 | -1141.5 | -1141.2 |
| | complete-U | -1148.3 | -1145.2 | -1146.8 | -1144.6 | -1150.7 | -1144.1 | -1140.3 | -1141.2 |
| | permuted | -1148.3 | -1146.6 | -1146.5 | -1144.6 | -1151.0 | -1144.5 | **-1140.0** | -1141.2 |
| hepatitis | standard IW | -561.3 | -774.9 | -775.4 | -774.4 | NaN | NaN | NaN | -1693.7 |
| | complete-U | **-561.2** | -564.3 | -773.1 | -774.4 | NaN | -1664.4 | **-1592.8** | -1693.7 |
| | permuted | -561.2 | -773.8 | -775.2 | -774.4 | NaN | -1779.6 | -1715.7 | -1682.1 |
| hiv-chr | standard IW | -606.4 | -604.4 | -607.5 | -603.8 | NaN | NaN | -1879.9 | -1945.0 |
| | complete-U | -606.2 | -604.0 | **-602.6** | -603.8 | NaN | -3395.8 | **-1450.6** | -1945.0 |
| | permuted | -606.2 | -604.0 | -603.1 | -603.7 | NaN | NaN | -1486.9 | -1960.9 |
| ionosphere | standard IW | -133.1 | -129.2 | -127.2 | **-125.6** | -125.3 | -126.6 | -132.5 | -142.3 |
| | complete-U | -133.2 | -129.6 | -127.3 | -125.6 | -124.9 | -125.2 | -127.2 | -142.3 |
| | permuted | -133.3 | -129.6 | -127.3 | -125.7 | -124.9 | -125.2 | -127.3 | -142.2 |
| irt | standard IW | -15887.5 | -15887.1 | -15886.8 | -15886.7 | -36563 | -64934 | -68447 | NaN |
| | complete-U | -15887.4 | -15887.0 | -15886.8 | -15886.6 | **-33230** | -37383 | -50316 | NaN |
| | permuted | -15887.4 | -15887.0 | -15886.8 | **-15886.6** | -35620 | -37547 | -54763 | NaN |
| irt-multilevel | standard IW | -15204.7 | -15194.1 | -15191.3 | -15196.8 | NaN | NaN | NaN | NaN |
| | complete-U | -15198.7 | -15164.0 | **-15185.8** | -15196.8 | NaN | NaN | NaN | NaN |
| | permuted | -15200.3 | -15173.0 | -15186.2 | -15197.0 | NaN | NaN | NaN | NaN |
| mesquite | standard IW | -29.9 | -29.7 | -29.6 | -29.3 | -29.8 | -29.7 | -29.6 | **-29.2** |
| | complete-U | -29.9 | -29.8 | -29.6 | -29.3 | -29.8 | -29.7 | -29.6 | -29.2 |
| | permuted | -29.9 | -29.7 | -29.6 | **-29.2** | -29.8 | -29.7 | -29.6 | -29.2 |
| mushrooms | standard IW | -211.6 | -206.5 | -204.3 | -215.5 | -180.8 | -194.2 | -215.8 | -339.0 |
| | complete-U | -210.6 | -204.3 | **-200.8** | -215.5 | **-180.2** | -180.7 | -185.6 | -339.0 |
| | permuted | -210.8 | -204.5 | -201.4 | -215.4 | -180.2 | -180.7 | -187.6 | -337.3 |
| radon | standard IW | -1210.5 | -1210.4 | -1213.3 | -1210.4 | NaN | -2422.9 | -1595.8 | -1636.5 |
| | complete-U | -1210.5 | -1210.2 | **-1210.2** | -1210.4 | NaN | -1548.0 | **-1445.9** | -1636.5 |
| | permuted | -1210.5 | -1210.2 | -1211.8 | -1210.4 | NaN | -1600.6 | -1454.7 | -1645.2 |
| sonar | standard IW | -136.2 | -126.3 | -121.3 | **-117.9** | -138.0 | -154.9 | -200.9 | -226.7 |
| | complete-U | -136.5 | -127.4 | -121.5 | -117.9 | **-116.6** | -120.9 | -156.3 | -226.7 |
| | permuted | -136.5 | -127.3 | -121.5 | -117.9 | -116.7 | -121.5 | -158.2 | -228.5 |
| wells | standard IW | -2042.1 | -2041.9 | -2041.7 | **-2041.2** | -2041.9 | -2041.8 | -2041.7 | **-2041.1** |
| | complete-U | -2042.2 | -2042.0 | -2041.7 | -2041.2 | -2041.9 | -2041.9 | -2041.8 | -2041.1 |
| | permuted | -2042.2 | -2041.9 | -2041.7 | -2041.2 | -2041.9 | -2041.8 | -2041.8 | -2041.1 |

Table 9: Median objective averaged over the last 200 iterations when using the complete-U DReG, permuted DReG and the standard DReG gradient estimators. It can be seen that for at least 8 models out of 17, using either the diagonal Gaussian or the full rank covariance Gaussian approximation, the best objective is achieved with an intermediate value of $m$, and it is at least 1 nat larger than the objective with $m = 16$. These models are: `congress`, `election88`, `election88Exp`, `electric`, `electric-one-pred`, `irt-multilevel`, `mushrooms` and `radon`. Optimizations using $m = 1$ are not shown.

| model | method | Diagonal Gaussian | | | | Full Rank Covariance Gaussian | | | |
|---|---|---|---|---|---|---|---|---|---|
| | | m | | | | | | | |
| | | 2 | 4 | 8 | 16 | 2 | 4 | 8 | 16 |
| a1a | DReG | -652.7 | -649.9 | -648.0 | -647.0 | -659.7 | -770.3 | -936.6 | -1205.4 |
| | comp.-DReG | -652.5 | -648.6 | -646.5 | -647.0 | -655.7 | -667.4 | -894.2 | -1205.4 |
| | perm.-DReG | -652.5 | -648.7 | **-646.4** | -647.1 | **-655.3** | -725.2 | -874.7 | -1209.8 |
| australian | DReG | -264.4 | -261.2 | -259.0 | **-257.8** | -256.7 | -256.7 | -256.7 | -256.9 |
| | comp.-DReG | -264.7 | -261.6 | -259.0 | -257.8 | -256.7 | -256.7 | **-256.6** | -256.9 |
| | perm.-DReG | -264.7 | -261.5 | -258.9 | -257.8 | -256.7 | -256.7 | -256.6 | -256.9 |
| congress | DReG | 417.9 | 419.7 | 419.9 | 418.2 | 418.5 | 418.9 | 413.2 | 404.5 |
| | comp.-DReG | 419.6 | 420.5 | 420.7 | 418.2 | 420.5 | **420.8** | 419.8 | 404.5 |
| | perm.-DReG | 419.4 | 420.5 | **420.7** | 417.9 | 420.4 | 420.7 | 419.8 | 404.6 |
| election88 | DReG | -1529.2 | -1522.3 | -1524.2 | -1534.9 | NaN | -5964.4 | -4349.2 | NaN |
| | comp.-DReG | -1529.1 | -1520.7 | **-1518.4** | -1534.9 | NaN | -4950.7 | **-2079.0** | NaN |
| | perm.-DReG | -1529.1 | -1520.7 | -1518.4 | -1534.3 | NaN | -5439.8 | -3041.2 | NaN |
| election88Exp | DReG | -1755.8 | -1571.9 | -1502.0 | -1461.9 | NaN | NaN | NaN | NaN |
| | comp.-DReG | -1733.2 | -1495.9 | -1468.8 | -1461.9 | NaN | NaN | **-3664.0** | NaN |
| | perm.-DReG | -1766.8 | -1512.3 | -1483.3 | **-1460.1** | NaN | NaN | -3947.5 | NaN |
| electric | DReG | -830.0 | -827.2 | -824.8 | -826.2 | NaN | -1417.4 | -1291.3 | -1314.0 |
| | comp.-DReG | -830.2 | -826.1 | **-822.0** | -826.2 | NaN | -1459.2 | **-1219.0** | -1314.0 |
| | perm.-DReG | -829.9 | -826.3 | -822.6 | -826.3 | NaN | -1427.2 | -1239.1 | -1326.1 |
| electric-one-pred | DReG | -1148.8 | -1147.5 | -1146.4 | **-1144.6** | -1153.0 | -1145.8 | -1141.4 | -1141.2 |
| | comp.-DReG | -1148.5 | -1146.0 | -1146.8 | -1144.6 | -1150.7 | -1144.1 | -1140.3 | -1141.2 |
| | perm.-DReG | -1148.3 | -1146.7 | -1146.5 | -1144.6 | -1151.0 | -1144.5 | **-1140.0** | -1141.2 |
| hepatitis | DReG | -561.3 | -776.3 | -775.1 | -774.0 | NaN | NaN | NaN | NaN |
| | comp.-DReG | **-561.1** | -772.0 | -772.6 | -774.0 | NaN | NaN | NaN | NaN |
| | perm.-DReG | -561.3 | -773.5 | -774.8 | -774.0 | NaN | NaN | NaN | NaN |
| hiv-chr | DReG | -606.2 | -604.1 | -605.4 | -602.7 | NaN | NaN | NaN | NaN |
| | comp.-DReG | -606.1 | -603.7 | **-602.2** | -602.7 | NaN | NaN | NaN | NaN |
| | perm.-DReG | -606.1 | -603.6 | -602.9 | -602.7 | NaN | NaN | NaN | NaN |
| ionosphere | DReG | -133.1 | -129.3 | -127.1 | **-125.6** | -124.3 | -125.8 | -130.7 | -142.1 |
| | comp.-DReG | -133.2 | -129.6 | -127.2 | -125.6 | **-124.2** | -124.2 | -124.7 | -142.1 |
| | perm.-DReG | -133.3 | -129.6 | -127.2 | -125.6 | -124.2 | -124.3 | -125.7 | -142.0 |
| irt | DReG | -15887.3 | -15886.9 | -15886.6 | **-15886.3** | NaN | NaN | NaN | NaN |
| | comp.-DReG | -15887.3 | -15886.9 | -15886.5 | -15886.3 | NaN | NaN | NaN | NaN |
| | perm.-DReG | -15887.3 | -15886.9 | -15886.5 | -15886.3 | NaN | NaN | NaN | NaN |
| irt-multilevel | DReG | -15226.1 | -15199.4 | -15195.8 | -15224.4 | NaN | NaN | NaN | NaN |
| | comp.-DReG | -15206.9 | -15188.6 | **-15188.0** | -15224.4 | NaN | NaN | NaN | NaN |
| | perm.-DReG | -15214.0 | -15191.6 | -15188.4 | -15222.2 | NaN | NaN | NaN | NaN |
| mesquite | DReG | -29.9 | -29.8 | -29.6 | **-29.4** | -29.8 | -29.7 | -29.7 | **-29.3** |
| | comp.-DReG | -29.9 | -29.8 | -29.6 | -29.4 | -29.8 | -29.7 | -29.7 | -29.3 |
| | perm.-DReG | -29.9 | -29.8 | -29.6 | -29.4 | -29.8 | -29.7 | -29.7 | -29.3 |
| mushrooms | DReG | -211.6 | -206.6 | -204.7 | -215.9 | -192.2 | -251.2 | -305.6 | -405.3 |
| | comp.-DReG | -210.6 | -204.3 | **-201.1** | -215.9 | **-180.3** | -193.6 | -253.5 | -405.3 |
| | perm.-DReG | -210.7 | -204.4 | -201.5 | -215.4 | -180.4 | -194.3 | -250.9 | -400.8 |
| radon | DReG | -1210.5 | -1210.3 | -1219.9 | -1210.3 | NaN | -2410.8 | -1624.9 | -1650.9 |
| | comp.-DReG | -1210.4 | **-1210.1** | -1210.2 | -1210.3 | NaN | -1538.0 | **-1445.7** | -1650.9 |
| | perm.-DReG | -1210.4 | -1210.2 | -1212.5 | -1210.2 | NaN | -1593.3 | -1466.2 | -1642.4 |
| sonar | DReG | -136.2 | -126.2 | -121.1 | **-117.6** | -135.4 | -152.5 | -226.3 | -259.6 |
| | comp.-DReG | -136.5 | -127.2 | -121.5 | -117.6 | **-115.2** | -118.4 | -155.6 | -259.6 |
| | perm.-DReG | -136.5 | -127.3 | -121.3 | -117.6 | -115.3 | -118.9 | -156.4 | -260.5 |
| wells | DReG | -2042.2 | -2041.9 | -2041.8 | **-2041.2** | -2041.9 | -2041.9 | -2041.8 | **-2041.2** |
| | comp.-DReG | -2042.2 | -2042.0 | -2041.9 | -2041.2 | -2041.9 | -2041.9 | -2041.9 | -2041.2 |
| | perm.-DReG | -2042.2 | -2042.0 | -2041.8 | -2041.2 | -2041.9 | -2041.9 | -2041.9 | -2041.2 |

