# OpenReview forum: "U-Statistics for Importance-Weighted Variational Inference"
_TMLR — Accepted by TMLR_

### Review · Reviewer_7r3U · 2022-12-06

**Summary Of Contributions:**

The paper improves the bias-variance trade-off of IWVI by using U-statistics. The main idea is to carefully construct overlapped batches to compute individual estimate for average. Several non-trivial estimates based on this idea are introduced, including
1. The complete U-statistics IW-ELBO estimator, which is computationally intense but achieves optimal variance reduction;
2. The permutation-based incomplete U-statistics IW-ELBO estimator, which allows controlling the computation by a hyper-parameter $l$ and achieves variance reduction in between the optimal and an incomplete estimator based on random selection;
3. Two approximate complete U-statistics IW-ELBO estimator, which uses the log-sum-exp approximation to allow fast approximation of the complete U-statistics.

Notably, theoretical results for each of these estimators are obtained in terms of variance. Most of them have already shown the advantage (better bounds) by using the newly proposed estimators and some of them also gives practical guidance on selection of hyper-parameters such as $l$ based on Proposition 4.2.

A large suite of simulation for IWVI is performed to validate the practical improvement by using the proposed estimators.
Benchmark on the actual computation is also provided to show the disadvantage of the complete U-statistics IW-ELBO estimator, and the importance of reducing computation achieved in the other estimators.
Finally the proposed estimators are tested in IWAEs, resulting in improved lower bound of the marginal likelihood.

**Audience:**

Yes

**Broader Impact Concerns:**

No concern as this work is mostly theoretical

**Claims And Evidence:**

Yes

**Requested Changes:**

I think the paper could be accepted as it.

**Strengths And Weaknesses:**

## Strengths
- The paper is very well-written and easy to follow. Related works are well-cited and enough technical background is provided.
- The proposed method is simple but effective. The use of U-statistics is very elegant and could motivate similar application in other related estimators for variance reduction.
- The paper consists a nice mix of theoretical results and empirical evaluation to study the proposed estimators.
    - Important theoretical results are established for guaranteed improvement from the proposed estimators as well as some guides of hyper-parameter selection.
    - Empirical results are aligned with the theory and show practical improvement by using the proposed estimators.

## Weaknesses
- Only a few typos or presentation issues
    - Missing brackets ($[$, $]$) in a few expectations ($\mathbb{E}$) within the paragraphs between (5) and (6)
    - The statement in Proposition 3.2 is incomplete. The first sentence should end up with something like "... for some $\zeta_1, \zeta_m > 0$" (otherwise they are not defined).

## Questions
- I wonder how the complete U-statistics estimator is implemented in practice and if any vectorization is used to accelerate it instead of a loop-based implementation.
    - For example, if we denote the weights as $\mathbf{e}=[e_1, e_2, e_3, e_4]$ and let $m=2$, one could write the complete U-statistics estimator as $h(\mathbf{e}^\top \mathbf{M}) \mathbf{1}  / r$ where $\mathbf{M}$ is a $4 \times 6$ binary matrix whose columns are indicators for each subset (e.g. the first column is $[1; 1; 0; 0]$, the second column is $[1; 0; 1; 0]$, etc.) and $h(x) = \ln {1 \over m} x$ is applied column-wise. This might be effectively parallelized if the vector-matrix multiplication is done on GPUs.
    - Just throwing a random idea: Maybe this vector-matrix multiplication could be somehow approximated efficiently (say by some low-rank approximation or even random projection).

---

> ### Author Response · Authors · 2022-12-07
> **Response to Reviewer 7r3U**
>
> We thank the reviewer for their comments and appreciate their attention to detail.
> In the following, we answer the reviewer’s questions.
>
> ## Regarding Implementation
>
> Both for IWVAEs and IWVI, we can get a vector of log_weights `log_weights = [v₁,..., vₙ]`.
> Then, for each of the estimators, we build a (flattened) set of indexes.
> For instance, if $n=3$ and $m=2$, for the complete-U estimator, we get `indexes = (0, 1, 0, 2, 1, 2)`.
> Finally, we use PyTorch's `index_select` and `logsumexp`.:
>
> ```python
> sets = log_weights.index_select(0, indexes).reshape(-1, m)
> (sets.logsumexp(-1) - np.log(m)).mean()
> ```
>
> We found that this implementation leverages PyTorch's parallelism without creating a (sparse) matrix.
>
> About your 'random idea,' the approximations we present in Section 5 are efficient approximations of the complete-U estimator inspired by the PyTorch implementation described above.
>
> ## Regarding notation
>
> Thanks for pointing this out. $\zeta_1$ and $\zeta_m$ were defined in Definition 3.
> We added a textual reference to this in Proposition 3.2.
>
> In inline-math mode, to avoid clutter, we decided to write expectations without brackets if the expression is sufficiently simple.
> For instance, when the expression is a function applied to a random variable, which is what you noticed in section 3.

---

### Review · Reviewer_JAwY · 2022-12-18

**Summary Of Contributions:**

The authors propose a new general recipe to derive variational bounds, in the spirit of importance weighted variational inference (IWVI), introduced by Burda et al. (2016) in the Importance-Weighted Autoencoders (IWAEs) paper.

The idea is quite simple yet novel in this context: using U-statistics instead of standard averages. Concretely, this means that the variational bound will be evaluated many times on different overlapping batches of importance weights. The novelty being the fact that the batches can be overlapping, and are much more numerous than for standard IWVI bounds. The authors use the standard theory of U-statistics to show that these new bounds have a lower variance than standard IWVI bounds. The downside of their new bounds is that computing all possible combinations of batches can be computationally intensive. The authors propose several solutions to this problem: sampling fewer batches than the complete U-statistic would, or relying on the fact that the logsumexp function can be approximated by the max function.

In several practical situations (Bayesian models and IWAEs), these new objectives are slightly more accurate than standard IWVI.

**Audience:**

Yes

**Broader Impact Concerns:**

I have no particular concern about this work.

**Claims And Evidence:**

Yes

**Requested Changes:**

- Clarifying/correcting the few points mentioned in the "weaknesses" section

- There are a few points/references that would be interesting to discuss:
    - You mention "we are not aware of control variates specifically for IWVI". Control variates for IWVI were developed by Mnih and Rezende (ICML 2016, https://arxiv.org/abs/1602.06725) and revisited by Liévin et al. (NeurIPS 2020, https://arxiv.org/abs/2008.01998).
    - DReG was extended by Bauer and Mnih (ICML 2020, https://arxiv.org/abs/2101.11046)
    - It would be nice to discuss whether or not your ideas could be applied also to the reweighted wake sleep algorithm of Bornschein and Bengio (ICLR 2015, https://arxiv.org/abs/1406.2751) and its extensions (eg Dieng and Paisley, https://arxiv.org/abs/1906.05850, Le et al., https://proceedings.mlr.press/v115/le20a.html, Kim et al., https://proceedings.mlr.press/v108/kim20b.html)




**Strengths And Weaknesses:**

Strengths

The idea of using U-statistics in this context is quite natural and excellent. Overall, the new objectives are practical, and often more accurate than standard IWAE-style bounds. We can hope that this work can benefit to several researchers using such bounds.

The paper is generally well-narrated, and cites most of the relevant literature.

The authors are refreshingly honest about the fact that the improvements are often slim.

Weaknesses

As mentioned above, the empirical gains are often slim.

Although the paper reads well, a few things are a bit unclear:

1) The maximum function is nonsmooth, which may be problematic for gradient descent? Discussing this would be interesting.

2) Some of the maths deserve a bit more details, for instance:
    - In the beginning of Section 2, $x$ and $z$ are not properly defined, and we have no idea where they live
    - In Def 3.1, it would be helpful to briefly say why it only depends on $c$
    - It would be nice to add a proof of Prop 3.3 (perhaps in the appendix) for completeness. While Prop 3.2 is exactly Hoeffding's result, Prop 3.3 involves some straightforward steps that would still benefit from being spelled out
    - It is perhaps well-known, but I was not familiar with the notation $\binom{[n] }{m}$, maybe specifying what it means would be helful
    - The computation needed to obtain Eqn (8) could be detailed
    - Right after Eqn (8), you say "for a given permutation $\pi$, all sets in $S_\pi$ will be independent". I don't understand, given $\pi$, aren't all sets actually deterministic?
    - In Appendix B, the paragraph before the theorem is nice but explaining the rationale of the conditions a bit more would be helful. For instance, why is $\tilde{\mathcal{F}}$ the class of finitely supported distributions? I am not particularly familiar with Halmos's famous paper, but the link between your result and the way he phrases Theorem 5 is not that obvious.

3) The use of the "envelope" of the curve in Figure 2 and later is quite peculiar, but not uninteresting. Is this something common? If yes, maybe you could motivate it a bit more (it is quickly explained in page 11, while Fig 2 is page 8, which is not great for the flow), and add citations of other papers that use it?

Some statements are incomplete or not fully correct:

a) Page 1: "The most obvious downside is the increased computational cost (by a factor of $m$)". Depending on the context, this can be smaller than this. For IWAEs, computing the standard bound requires a single encoding, and $m$ decodings, whereas increasing the complexity by a factor m would mean $m$ encodings, and $m$ decodings.

b) Page 2: when you mention that the IWAE bound is consistant, you should mention that this is under some assumptions on the distribution of the importance weights (Burda et al. assume boundedness, Domke and Sheldon have weaker conditions)

c) Page 2:  "which simultaneously finds an approximating distribution that is close in KL divergence to p(z | x) (Domke & Sheldon, 2018)" I believe Domke & Sheldon actually show that, when $K \longrightarrow \infty$, the distributions are close in $\chi$ divergence instead of KL. I am not aware of precise results on what's happening for $K \in \]1,\infty[$.

---

> ### Author Response · Authors · 2023-01-06
> **Response to Reviewer JAwY  (part 1 of 2)**
>
> We thank the reviewer; your comments helped to improve our submission.
> We respond inline.
>
> ## Clarifications
>
> > 1. The maximum function is nonsmooth, which may be problematic for gradient descent? Discussing this would be interesting.
>
> - We use `sort` for the approximations, which shares some of the properties of `max`: if the distribution of the importance-sampling weights is absolutely continuous, then `sort` is differentiable with probability one (because the probability of two weights being equal is zero).  We included Note 5.4 clarifying this.
>
> > 2. Some of the maths deserve a bit more details, for instance:
> >    - In the beginning of Section 2, $x$ and $z$ are not properly defined, and we have no idea where they live
>
> We clarified that both $X$ and $Z$ are random vectors, and their relizations live, respectively, in $\mathbb R^{d_{\mathcal X}}$ and $\mathbb R^{d_{\mathcal Z}}$.
>
> >- In Def 3.1, it would be helpful to briefly say why it only depends on $c$
>
> - We found a typo in Def 3.1: It should have been $\binom{[2m]}{m}$ instead of $\binom{[n]}{m}$.
> The reason why the definition does not depend on the particular choices of $\mathbf{s}, \mathbf{s}' \in \binom{[2m]}{m}$ is because the $V_i$ are iid and $h$ is a symmetric function: the distribution of $h(V_{s_1}, \dots, V_{s_m})$ is the same as the distribution of $h(V_1, \dots, V_m)$; if $c = \lvert\mathbf{s} \cap \mathbf{s}’\rvert$, then the distribution of $h(V_{s_1}, \dots, V_{s_m})h(V_{s_1’}, \dots, V_{s_m’})$ is the same as that of $h(V_1, \dots, V_m)h(V_1, \dots, V_c, V_{m+1}, \dots, V_{2m-c})$.
> We added a sentence with the intuition of the definition: “In words, this is the covariance between two IW-ELBO estimates, each using one batch of $m$ i.i.d. log-weights, and where the two batches share c log-weights in common.”
>
> > - It would be nice to add a proof of Prop 3.3 (perhaps in the appendix) for completeness. While Prop 3.2 is exactly Hoeffding's result, Prop 3.3 involves some straightforward steps that would still benefit from being spelled out
>
> We included a small proof in Appendix B.1.
>
> > - It is perhaps well-known, but I was not familiar with the notation $\binom{[n]}{m}$ , maybe specifying what it means would be helpful.
>
> When $A$ is a set containing $n$ elements, it is relatively common in combinatorics to use $\binom{A}{m}$ to denote the set of all subsets of $A$ with exactly $m$ elements [see Sagan (2020)]. It has the nice property that $\bigcup_{m=0}^n \binom{A}{m} = 2^A$, where the right hand side is the (more common) way to denote the power-set of $A$.
>   - Sagan, Bruce E. *Combinatorics: The art of counting*
>             . Vol. 210. American Mathematical Soc., 2020.
>
> >- The computation needed to obtain Eqn (8) could be detailed.
>
> We updated the document with the detailed calculation.
>
> >- Right after Eqn (8), you say "for a given permutation $\pi$, all sets in  $\mathcal{S}_{\pi}$ will be independent". I don't understand, given , aren't all sets actually deterministic?
>
> We rephrased the proof to improve clarity.
>
> >- In Appendix B, the paragraph before the theorem is nice but explaining the rationale of the conditions a bit more would be helpful. For instance, why is  the class of finitely supported distributions? I am not particularly familiar with Halmos's famous paper, but the link between your result and the way he phrases Theorem 5 is not that obvious.
>
> We made the conditions in Proposition B.1  stricter than those from Halmos to simplify the exposition.
> Using Halmos's notation, we set $E = \mathbb{R}$, and $\mathcal D = \tilde{\mathcal F}$ to be the set of distributions finitely closed over the real line.
> We added the condition that the estimator $\Phi$ is unbiased for every distribution in $\tilde{\mathcal F}$ (which in Halmos, it is a consequence of Thm. 4).
> With all of that, we can apply Theorem 5.
>
> >3. The use of the "envelope" of the curve in Figure 2 and later is quite peculiar, but not uninteresting. Is this something common? If yes, maybe you could motivate it a bit more (it is quickly explained in page 11, while Fig 2 is page 8, which is not great for the flow), and add citations of other papers that use it?
>
> We appreciate your interest in that metric, which is an iteration over an idea that appeared in Geffner and Domke (2018).
> We added a citation to that paper and a small clarification.
>
> - Geffner, Tomas, and Justin Domke. *Using large ensembles of control variates for variational inference.* Advances in Neural Information Processing Systems 31 (2018).

---

> > ### Author Response · Authors · 2023-01-06
> > **Response to Reviewer JAwY (part 2 of 2)**
> >
> > ## Completion/Correction of statements.
> >
> > > a) Page 1: "The most obvious downside is the increased computational cost (by a factor of m)". Depending on the context, this can be smaller than this. For IWAEs, computing the standard bound requires a single encoding, and m decodings, whereas increasing the complexity by a factor m would mean m encodings, and m decodings.
> >
> > We changed the wording.
> >
> > > b) Page 2: when you mention that the IWAE bound is consistant, you should mention that this is under some assumptions on the distribution of the importance weights (Burda et al. assume boundedness, Domke and Sheldon have weaker conditions)
> >
> > You are right about the subtleties of the assumptions.
> > We included references to the sources concerning the assumptions.
> >
> > > c) Page 2: "which simultaneously finds an approximating distribution that is close in KL divergence to p(z | x) (Domke & Sheldon, 2018)" I believe Domke & Sheldon actually show that, when K⟶∞, the distributions are close in χ divergence instead of KL. I am not aware of precise results on what's happening for K∈]1,∞[.
> >
> > In our citation to the work of Domke & Sheldon (2018), we meant to refer to their Theorems 1 and 2, where the gap between $\ln p(x)$ and the IW-ELBO is shown to be equal to a KL divergence between augmented distributions.
> >
> > ## Related work
> >
> > We appreciate the references to related works provided by the reviewer.
> > These works show that our contribution could be applied more broadly than we initially thought, providing exciting directions for future research.
> >
> > ### Generalized Doubly-Reparameterized Gradient Estimators
> >
> > Since the DReGs estimator for hierarchical IWAE introduced in Bauer and Mnih (2021) uses self-normalizing weights, it faces the trade-off described in section 2.1 of our paper. Hence, we expect that it will benefit from using the techniques that we introduced. However, we left this analysis outside the scope of our paper.
> > We included a reference to Bauer and Mnih (2021) in the Related Work Section.
> >
> > - Matthias Bauer and Andriy Mnih. "Generalized Doubly Reparameterized Gradient Estimators." _International Conference on Machine Learning._ PMLR, 2021.
> >
> > ### Control Variates
> >
> > We appreciate the references provided by the reviewer. We now mention the works by Mih and Rezende (2016) and the extention in Liévin et al. (2020).
> > The use of control variates for IWVI was mainly focused on using the score function gradient estimator. In the future, it would be interesting to explore if the VIMCO of Mih and Rezende (2016) or OVIS introduced by Liévin et al. (2020) will benefit from using incomplete U-statistics.
> > The theoretical result of Liévin et al. (2020) is similar to that of Tucker et al. (2018), i.e., that the SNR grows as $\mathcal{O}(m^{1/2})$, but for the score function gradient estimator. Interestingly, that asymptotic result was derived when $m\to \infty$, without characterizing the behavior of $n$. Hence, the trade-off between $m$ and $n$ was not explicit.
> > Since the ideas presented in our paper improved the performance of the DReG estimator, it wouldn't be surprising that the same applies to the OVIS estimator.
> >
> > - Andriy Mnih and Danilo Rezende. "Variational inference for monte carlo objectives." _International Conference on Machine Learning._ PMLR, 2016.
> > - Liévin, Valentin, et al. "Optimal Variance Control of the Score-Function Gradient Estimator for Importance-Weighted Bounds." _Advances in Neural Information Processing Systems_ 33 (2020): 16591-16602.
> > - Tucker, George, et al. "Doubly Reparameterized Gradient Estimators for Monte Carlo Objectives." _International Conference on Learning Representations._ 2018.
> >
> > ### Reweighted Wake-Sleep
> >
> > As the reviewer noticed, variations of the Reweighted Wake-Seep algorithm [citations below] could be improved by using a complete- or incomplete-U-statistic. And this is especially true for the Wake-phase.
> > We now mention this in the Future Work section with references to Bornschein and Bengio (2015), Dieng and Paisley (2019), Le et al. (2020), and Kim, Hwang, and  Kim (2020).
> >
> > - Jörg Bornschein and Yoshua Bengio. Reweighted wake- sleep. In International Conference on Learning Repre- sentations, 2015
> > - Adji B. Dieng and John Paisley. "Reweighted expectation maximization." arXiv preprint arXiv:1906.05850 (2019).
> > - Tuan Anh Le et al. "Revisiting reweighted wake-sleep for models with stochastic control flow." Uncertainty in Artificial Intelligence. PMLR, 2020.
> > - Dongha Kim, Jaesung Hwang, and Yongdai Kim. "On casting importance weighted autoencoder to an EM algorithm to learn deep generative models." International Conference on Artificial Intelligence and Statistics. PMLR, 2020.

---

### Review · Reviewer_j1dR · 2023-01-10

**Summary Of Contributions:**

This paper applies the theory of U-statistics to multisample bound and gradient estimators for variational inference. In many variational inference settings the user allots a budget of samples that can be drawn from the model at each gradient step. Those samples are then combined to estimate the bound and its gradient. The most popular existing method for combining the samples is, roughly speaking, to average each sample's log importance weight (IWAE). This paper suggests using the theory of U-statistics to more intelligently combine the samples and achieve a variance reduction. This can take the form of averaging multiple possibly-overlapping subsets of the weights.

In addition to these conceptual innovations, the authors propose several estimators based on U-statistics and analyze their performance theoretically and empirically, finding that they match or exceed existing estimators while incurring a small performance penalty.

**Audience:**

Yes

**Broader Impact Concerns:**

No broader impact concerns.

**Claims And Evidence:**

Yes

**Requested Changes:**

I recommend acceptance of this paper without needing any changes. However, there are a few things that would be nice to see. Mainly, it would be nice to add a discussion of batching vs averaging multiple IWAE bounds for the same x.

In addition here are a few small optional edits:

* On the top of page 3 you say “we use the subscript n to denote the total number of input samples used for estimation and m for the optimized IW-ELBO objective”. Although I was able to figure it out, it's not really clear to me from this sentence what m is being used for.
* On the bottom of page 4 you say "can be generalized by replacing h by any other symmetric function of m variables”. While 'symmetric function of m variables' does have the technical definition 'invariant to permutations of the variables', I had to look it up just to make sure. It could be better to be more precise here, but it is probably fine as-is.
* In Definition 3.1 you end saying “which depends only on c and not the particular s and s'” It kind of sounds like you are making a claim, not just a definition when you add the above statement. Maybe move it out of the definition?
* It could be good to show table 2 for different values of n and m to support your claims in the text. The additional info in the appendix for the KMNIST experiments kind of gets at this this, but it could be good to support your claims more directly with experiments on the mushroom dataset.


**Strengths And Weaknesses:**

Strengths:
* The paper is clear and well written.
* The contributions are novel, interesting, and useful.
* I checked the proofs in the main text and they are correct to my knowledge.
* The experimental evaluation is thorough and provides clear evidence to support the proposed methods.
* Overall it is a great paper, congratulations!

The main weakness I see is a lack of discussion of 'outer averaging' vs batch averaging for variance reduction. To my understanding, the standard IWAE setup does not do any outer averaging as defined in this paper. Instead, all averaging occurs at the 'batch level', i.e. IWAE bounds for different x's are averaged, not multiple IWAE bounds for the same x. I understand that your section 2.1 is largely dedicated to arguing that practitioners should consider averaging multiple bounds for the same x, but you ignore the option of instead averaging IWAE bounds for different $x$'s. This is an important and practical baseline to consider.

---

> ### Author Response · Authors · 2023-01-12
> **Response to Reviewer j1dR**
>
> We thank the reviewer for their comments and suggestions.
> We introduced some of the suggestions into the paper and responded to their points here.
>
> ## Outer averaging vs Batch averaging
>
> As noted by the reviewer, the batch size is an extra knob that might impact the estimation variance and hence the optimization's speed and quality.
> The objective of this paper is to show that if the user decides to use $n=rm$ samples for every $x$, with $r > 1$ and $m >1$, she will be better off using the estimators presented in this paper.
> In that sense, using $r> 1$ or increasing the batch size could be considered in the hyper-parameters optimization phase.
> Lastly, notice that the setting with $r> 1$ is, for instance, the one used by Rainforth et al. (2018): they used the letter $M$ to denote what we called $r$ and $K$ for what we called $m$.
>
> - Tom Rainforth, et al. "Tighter variational bounds are not necessarily better." _International Conference on Machine Learning._ PMLR, 2018.
>
> ## "Optional Edits"
>
> ### Definition of $m$
>
> Thank you for the suggestion.
> We reworded the definition of $m$ to highlight that it is the number of arguments of $h$ and that it determines the IW-ELBO objective to be optimized.
>
> ### Symmetric functions
>
> Thank you for pointing out this.
> We included a footnote clarifying the definition of a _symmetric function_.
>
> ### Definition 3.1
>
> You are right regarding the last sentence of Definition 3.1: it is a claim.
> We included that sentence in the Definition because (1) it proves that the $\zeta_c$ are well-defined, and (2)) to avoid creating clutter with minor propositions.
>
> ### Results in Table 2
>
> We appreciate your comment regarding the experimental results in Table 2.
> We chose not to delve too deeply into timing experiments because: (1) there are inherent difficulties in accurately measuring wall-clock performance and (2) the amount of extra computation for U-statistic methods is easy to predict in advance from known problem parameters.
> For example, the complete U-statistic scales linearly with $\binom{n}{m}$, and for the approximation, it is the time for one sort.
> The critical conclusion of Table 2 is that, in one simple setting, this extra computation can be quite modest compared to other model computations.
> We observed informally this is also true in many other settings.
> For a related timing experiment, see our Jackknife experiments in Appendix A.

---

### Decision · Action_Editors · 2023-02-23

**Recommendation:** Accept as is

**Comment:**

All reviewers were very supportive of the paper and recommended its acceptance without revision.  I concur with this assessment and believe it to be a clearly presented paper with interesting and notable novel contributions to the literature.

**Audience:**

Everyone agrees that the paper will be of interest to the community.  There is clear novelty in the contributions, the work focus on an area of research that is will of interest to many people in the TMLR community, and the paper itself situates the contributions well in the context of existing literature.

**Claims And Evidence:**

All the reviewers and I agree that the claims of the work are accurate and well-evidence by both theoretical and empirical results.  The authors are honest about the limitations of the work, especially the fact that the empirical improvements are often relatively small.